# Reshape and Adapt for Output Quantization (RAOQ): Quantization-aware Training for In-memory Computing Systems

## Abstract

In-memory computing (IMC) has emerged as a promising solution to address both the computation and data-movement challenges posed by modern AI models. IMC takes advantage of the intrinsic parallelism of memory hardware and performs computation on data in-place directly in the memory array. To do this, IMC typically relies on analog operation, which enables high energy and area efficiency. However, analog operation makes analog-to-digital converters (ADCs) necessary, for converting results back to the digital domain. This introduces an important new source of quantization error, impacting inference accuracy. This paper proposes a Reshape and Adapt for Output Quantization (RAOQ) approach to overcome this issue, which comprises two classes of mechanisms motivated by the fundamental impact and constraints of ADC quantization, including: 1) mitigating ADC quantization error by adjusting the statistics of activations and weights, through an activation-shifting approach (A-shift) and a weight reshaping technique (W-reshape); 2) adapting AI models to better tolerate ADC quantization, through a bit augmentation method (BitAug) to aid SGD-based optimization. RAOQ demonstrates consistently high performance across different scales of neural network models for image classification, object detection, and natural language processing (NLP) tasks at various bit precisions, achieving state-of-the-art accuracy with practical IMC implementations.

## 1 Introduction

Rapid advances in AI have greatly impacted various application domains, including computer vision, natural language processing, speech, etc. Recent generative AI breakthroughs have pushed the strength of AI even further, producing remarkably realistic and imaginative outputs, blurring the line between human- and machine-generated content (OpenAI, 2023; Chowdhery et al., 2022). However, increasing AI capability has come from increasing model complexity, with a sharp rise in both the number of compute operations and the number of model parameters, placing huge demands on hardware resources (Villalobos et al., 2022; Smith et al., 2022).

This has driven the development of specialized hardware architectures to accelerate AI model computations. While digital accelerators have been widely deployed to improve compute efficiency, they do not address the large amount of data movement involved, which has been shown to pose a critical energy and performance bottleneck in state-of-the-art (SOTA) models (Verma et al., 2019). In-memory computing (IMC), on the other hand, performs computations in place on stored data, providing an approach to simultaneously address both compute efficiency and data movement.

While both digital and analog IMC have been proposed, providing various advantages and trade-offs towards energy efficiency and accuracy, this work focuses on energy-aggressive highly-parallel analog IMC, addressing the critical bottleneck within the architecture via algorithmic solutions. A fundamental requirement of analog IMC is the need for analog-to-digital converters (ADCs), to provide compute outputs back to the digital domain for further processing. Importantly, ADCs introduce an additional source of quantization, which can substantially degrade accuracy in SOTA AI models. The level of quantization error from the ADC is fundamentally set by the level of IMC parallelism, which also directly sets the compute efficiency and throughput advantage.

Unlike quantization of activations and weights, whose clipping parameters can be directly optimized during training, ADC quantization on the compute results does not provide this degree of freedom and thus requires new methods to address. Previous works introduce artificial clipping to model ADC quantization at the hardware design stage (Gonugondla et al., 2020; Sakr & Shanbhag, 2021). However, this limits hardware flexibility in supporting various types of models, which may present different ADC-input data distributions and thus require different optimal clipping values.

To address such quantization challenges, this paper presents Reshape and Adapt for Output Quantization (RAOQ), to tackle the challenges at the algorithmic level. As neural networks generally are sensitive to drastic changes, we first perform quantization-aware training (QAT) for activations and weights only, and then apply RAOQ, in another stage of training with ADC quantization introduced. We explore RAOQ across multiple applications, i.e., image classification, object detection, and natural language processing (NLP), on ImageNet (Deng et al., 2009), COCO 2017 (Lin et al., 2014), and SQuAD 1.1 (Rajpurkar et al., 2016) datasets, respectively. To the best of our knowledge, this work is the first to demonstrate approaches that enable IMC for inference across various scales of models and challenging datasets/tasks. The major contributions of our work are as follows:

1. We conduct an analysis of the relationship between neural network activations, weights, and ADC quantization. We identify the statistical attributes of activations and weights that yield a high signal-to-quantization-noise ratio (SQNR) in the presence of ADC quantization.

2. We propose an activation-shifting method (A-shift) motivated by the preferred statistical attributes for activations, and a weight-reshaping technique via kurtosis regularization (W-shape) motivated by the preferred statistical attributes for weights.

3. We propose bit augmentation (BitAug), where the model is augmented in the dimension of ADC bit precision to aid the optimization process, assisting model adaptation to ADC quantization.

4. We conduct experiments on different models and tasks (i.e., ReNet18/50 (He et al., 2016), MobileNetV2 (Sandler et al., 2018), EfficientNet-lite0 (Tan & Le, 2019), YOLOv5s (Jocher et al., 2022), BERT-base/large (Devlin et al., 2018)), and across different quantization levels for activations, weights, and ADCs. The consistently high performance achieved by our proposed methods provides promise for their generalizability across challenging AI tasks.

## 2 BACKGROUND AND RELATED WORKS

### 2.1 IN-MEMORY COMPUTING (IMC)

IMC aims to address both compute and data-movement costs in matrix-vector multiplications (MVMs), which are dominant operations in modern AI models. This is achieved by storing matrix weights in a 2D array of memory bit cells as shown in Fig. 1a, and accessing compute results over many weight bits, rather than accessing the individual weight bits themselves. Specifically, this is achieved by performing multiplication in each bit cell between stored weight data and provided input data, and then accumulation to reduce the products in each column to a single compute result. The level of reduction, set by the row parallelism of IMC operation, thus determines the energy efficiency and throughput gains.

To enable energy- and area-efficient computation within the constrained bit cells, IMC can leverage analog operation, where the compute results then need to be converted back to the digital domain via ADCs (Valavi et al., 2019; Lee et al., 2021b; Deaville et al., 2022; Yin et al., 2020; Hsieh et al., 2023). Such analog operation raises two challenges. First, it is sensitive to noise sources, which degrade the output signal-to-noise ratio (SNR). Researchers have proposed algorithmic noise-aware training approaches to overcome this (Zhang et al., 2022; He et al., 2019), but which have only shown success in simple tasks (MNIST, CIFAR-10/100 datasets) at low levels of IMC row parallelism. Instead, recent work has moved to a high-SNR form of IMC, overcoming such analog noise, enabling scale-up to higher levels of row parallelism (Jia et al., 2022; Lee et al., 2021a). This has left SOTA analog IMC primarily subject to the second challenge, which is ADC quantization. As an example, Fig. 1b shows the degraded SQNR due to ADC quantization and inference accuracy in ResNet50 on ImageNet. Consequently, such quantization prevents IMC from scaling up and poses an ultimate limitation to the IMC efficiency and throughput. While ADC precision can be increased for higher SQNR and accuracy, this brings substantial hardware cost, with ADCs showing dominating energy

consumption (Lee et al., 2021a). This work introduces efficient algorithmic approaches to address the critical challenge in IMC systems today, which is ADC quantization, doing so without incurring additional hardware costs, to demonstrate applicability on a critical set of models.

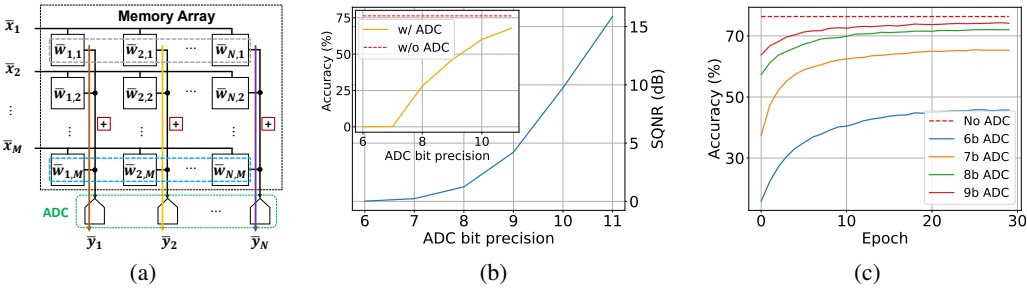

Figure 1: (a) An illustration of an MVM operation via IMC. (b) SQNR and accuracy degradation due to ADC quantization. (c) Learning curves for conventional QAT with ADC quantization involved.

## 2.2 QUANTIZATION-AWARE TRAINING (QAT)

QAT restores model accuracy, which may otherwise degrade due to quantization noise, through a training process that adapts the model parameters. QAT methods have been proposed to successfully demonstrate SOTA accuracy in aggressively quantized networks (Jacob et al., 2017; Gupta et al., 2015; Louizos et al., 2018; Bhalgat et al., 2020; Jain et al., 2019; Zhou et al., 2016; Nagel et al., 2022; Wang et al., 2022; Park et al., 2022; Esser et al., 2019). However, previous QAT mainly focuses on quantization from inputs (i.e., weight and activation), not considering ADC quantization on the compute outputs in IMC. As a result, IMC shows substantially degraded model accuracy even with conventional QAT, as seen in Fig. 1c.

To address ADC quantization in IMC, Jin et al. (2022) introduces a modified straight-through estimator (STE) (Bengio et al., 2013) along with calibration and rescaling techniques to assist the QAT process, demonstrating ResNet models on CIFAR-10/100 datasets. Sun et al. (2021) proposes a non-uniform activation quantization scheme and a reduced quantization range, validating on the CIFAR-10 dataset. Wei et al. (2020) proposes modified minmax quantizers for activations and weights to incorporate hardware statistics of IMC, testing on MNIST and CIFAR-10 datasets. While these prior works show success on relatively simple datasets, their success has not transferred to more complicated datasets and AI tasks. In this work, we propose improved QAT techniques to enable SOTA accuracy applicable to various bit precisions on more challenging models and tasks.

## 3 ANALYSIS AND RATIONALE FROM ADC QUANTIZATION

To formally define the IMC ADC quantization problem, let $x \in \mathbb{R}^M$ be a data vector of the activation $X$ and let $w \in \mathbb{R}^M$ be a vector of an output channel of the weight $W$. Denote $\overline{x}$ and $\overline{w}$ as their quantized counterparts, an IMC column then computes a portion of MVM:

$$y = <\overline{x}, \overline{w}> = \sum_{i=1}^{M} \overline{w}_i \overline{x}_i. \tag{1}$$

Note that convolutions can be converted to MVMs via *im2col* operations. For a $b_x$-bit activation, $b_w$-bit weight, $b_a$-bit ADC, and memory with dimension $M \times N$, assuming symmetric quantization is applied to weights, the ADC quantization and its quantization step $\Delta_a$ is defined as

$$\overline{y} = \left\lfloor clip\left(\frac{y}{\Delta_a}, n_a, p_a\right) \right\rfloor \tag{2}$$

$$\Delta_a = \frac{2M(2^{b_x} - 1)(2^{b_w-1} - 1)}{2^{b_a} k}, \tag{3}$$

where $\lfloor \cdot \rfloor$ denotes the floor operation. Similar to conventional QAT, the gradient of the floor operation is approximated using STE (Bengio et al., 2013). Above, $(n_a, p_a) = (-2^{b_a-1}, 2^{b_a-1} - 1)$, and

$k$ is a positive integer, serving as a hardware design parameter to provide fixed clipping (due to the ADC's supported input range). Eq. 3 assumes unsigned activations. For signed activations, we can simply replace $2^{b_x} - 1$ by $2^{b_x - 1} - 1$. In general, $\Delta_a$ is fixed for given hardware and is not trainable at the algorithmic level. Fig. 2a shows the distribution of an ADC input from ImageNet dataset via the ResNet50 model. We see that the input concentrates around a small portion of the ADC range, resulting in a small signal, relative to the quantization step $\Delta_a$. A choice of large $k$ could help to have a finer step $\Delta_a$, but would potentially introduce substantial clipping error. As different layers and models lead to different statistics of the compute outputs (ADC inputs), there is no optimal $\Delta_a$ to rule them all. Thus, with no algorithmically controllable parameters for ADC quantization, the only degrees of freedom left are parameters applicable to the activations and weights.

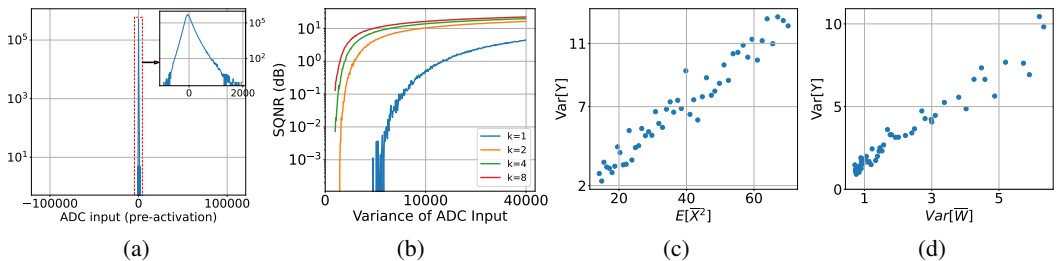

Figure 2: (a) Distributions of ADC input (compute output). (b) Relationship between ADC SQNR and the variance of ADC input $Var\left[Y\right]$. (c-d) Relationship of the variance of ADC input to the $2^{nd}$ moment of the quantized activation and the variance of the quantized weight.

Based on observations in Fig. 2a, we prefer the variance of the ADC input $Var\left[Y\right]$ to be maximized in order to maximize signal power and utilize as many ADC quantization levels as possible. This is explicitly shown in Fig. 2b, via the post-ADC SQNR. This focus on $2^{nd}$-order statistics, makes it natural to consider the dependence on the $2^{nd}$ moment of the activation $X$ and weight $W$. Before the training starts, activations and weights are independent of each other, and $Var\left[Y\right]$ is proportionally set by $\mathbb{E}[\overline{X}^2]$ and $\mathbb{E}[\overline{W}^2]$. However, during and after training, generally the assumption of independence does not hold, as $X$ and $W$ exhibit correlation through the neural network learning process. Nonetheless, we postulate that a more narrow relationship holds, namely that there is direct dependence between the $2^{nd}$ moments, and we conduct an empirical study to validate this. We randomly sample images from CIFAR10 and ImageNet datasets, and also randomly generate input data. We use ResNet50 and MobileNetV2 as example networks to perform standard QAT, since these contain the network structures encountered in most SOTA models. To manage computation complexity, we only take the first few layers of these models for this study. In Fig. 2c-2d, we plot the variance of the ADC input $Var[Y]$ vs. $\mathbb{E}[\overline{X}^2]$ and $\mathbb{E}[\overline{W}^2]$, respectively, and observe a proportional relationship. Further, since neural network weights are typically symmetrically distributed around zero (Bhalgat et al., 2020), $\mathbb{E}[\overline{W}^2]$ can be taken to be $Var[\overline{W}]$, and we postulate that $Var\left[Y\right]$ can be increased by maximizing $Var[\overline{W}]$ and $\mathbb{E}[\overline{X}^2]$, to improve IMC SQNR in the presence of ADC quantization. This rationale forms the basis of the W-reshape and A-shift techniques that form the proposed ROAQ approach described below. In the following sections, we use $\mathcal{L}_Q$ to denote the loss during the QAT stage of training and use $\mathcal{L}_A$ to denote the loss during the RAOQ stage of training, after QAT.

## 4 RESHAPE AND ADAPT FOR OUTPUT QUANTIZATION (RAOQ)

### 4.1 SQNR ENHANCEMENT

**Weight reshaping (W-reshape).** To maximize $Var[\overline{W}]$, one option is to perform aggressive scaling during quantization. However, this is expected to introduce substantial clipping error, posing an adverse trade-off with weight distortion. Thus, we seek an alternate approach to increasing $Var[\overline{W}]$, by adapting the distribution shape to avoid severe clipping.

Neural network weights typically exhibit a symmetric distribution in the exponential family, e.g., normal distribution or Laplace distribution (Banner et al., 2019; Shkolnik et al., 2020), which results

in relatively low variance. We therefore propose a penalty on weights to drive towards a distribution with a large variance, in a manner where the penalty does not degrade previous accuracy. This is achieved by introducing a kurtosis loss as a function of the quantized weights. Kurtosis describes the tailedness of a distribution, and such loss is defined as the standardized $4^{th}$ moment, i.e.,

$$\kappa = \mathbb{E}\left[\left(\frac{\overline{W} - \mu_{\overline{W}}}{\sigma_{\overline{W}}}\right)^4\right],\tag{4}$$

where $\overline{W}$ is the quantized weight, $\mu_{\overline{W}}$ and $\sigma_{\overline{W}}$ denote the mean and standard deviation of $\overline{W}$.

This encourages the majority of $\overline{W}$ to be concentrated in the tails of the distribution (Moors, 1986). This is different from (Shkolnik et al., 2020), where kurtosis loss is applied on the floating-point weights specifically to drive them towards a uniform distribution, which maximizes their quantization robustness. Since our interest is in improving ADC quantization, rather than weight quantization, we apply more aggressive kurtosis loss on the already quantized weights, which are determined by both the statistics of the floating point weights and the quantization parameters. We analyze the impact of W-reshape on the QAT accuracy and provide details in Appendix A. This loss, computed at each layer, is summed up to produce the final loss and then combined with the original loss function $\mathcal{L}_c$ against the ground truth during the QAT stage, i.e.,

$$\mathcal{L}_Q = \mathcal{L}_c + \lambda_\kappa \sum_l \kappa_l,\tag{5}$$

where $\lambda_\kappa$ is a coefficient to balance the two loss terms, and $l$ is an index for neural network layers. Fig. 3a (top) shows a comparison between the quantized weight with and without incorporating the kurtosis loss. We can see that the proposed method successfully reshapes the weight distribution to have a much larger variance, i.e., $4\times$ more than the case without $\mathcal{L}_\kappa$.

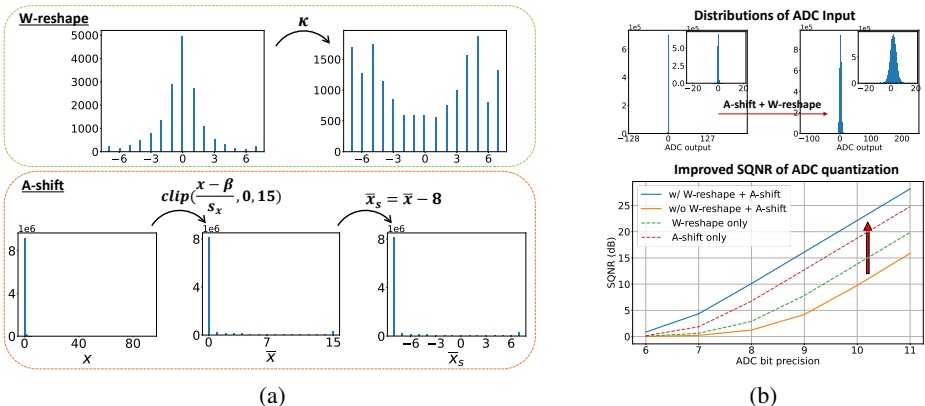

(a)                                                                      (b)

Figure 3: (a) Demonstration of W-reshape and A-shift for 4b weights and activations. (b) SQNR improvement under ADC quantization. This can be directly observed in terms of the utilization of the ADC intervals. The proposed techniques provide nearly $5\times$ utilization improvement.

**Activation shifting (A-shift)** In order to maximize the $2^{nd}$ moment of the activation, it is desirable for activations to exhibit a concentration of mass at considerably large absolute values, i.e., distancing the mass from zero, so that the input distribution to the ADC has maximum variance. However, this is typically not the case with activations derived from functions like SiLU (Elfwing et al., 2018) and GELU (Hendrycks & Gimpel, 2016), which inherently exhibit significant mass distribution around small values in close proximity to zero, as shown in Fig. 3a (bottom left).

Exploiting the fact that quantizing these activations as a signed number or unsigned number does not have much impact on the overall performance (Bhalgat et al., 2020), we propose to treat them as an unsigned number during quantization, and then convert them to a signed number. This yields a distribution moved away from zero, to the advantage of ADC quantization. Such an unsigned-to-signed conversion can be implemented by a simple shift:

$$\overline{x}_s = \left\lfloor clip\left(\frac{x - \beta}{s_x}, 0, 2^{b_x} - 1\right)\right\rceil - 2^{b_x - 1} = \overline{x} - 2^{b_x - 1}\tag{6}$$

where $\lfloor \cdot \rceil$ denotes round operation, $\beta$ and $s_x$ are trainable quantization parameters. Fig. 3a (bottom) shows the entire A-shift process. We observe that the mass of $\overline{x}_s$ is concentrated at the most negative values, hence having an extremely large $2^{nd}$ moment. On the contrary, quantizing activations directly to a signed number prevents such a shift operation, resulting in a much smaller $2^{nd}$ moment. To quantitatively verify our arguments, we compute the numerical values of the $2^{nd}$ moment for the quantized activation from the proposed method and from signed quantization based on Fig. 3a, ending up with 57.9 and 3.89, respectively. Our proposed approach produces a much greater $2^{nd}$ moment, roughly $15\times$ higher. Additionally, ReLU activation functions naturally suit the A-shift approach, as they explicitly force the output activations to be unsigned numbers. With such shifting, the IMC computation becomes

$$y = \sum_{i=1}^{M} \overline{w}_i \overline{x}_i = \sum_{i=1}^{M} \overline{w}_i \overline{x}_{s,i} + \underbrace{2^{bx-1}\overline{w}_i}_{\text{offset}} \tag{7}$$

The additional offset introduced by A-shift can be precomputed offline and thus does not add any overhead when performing inference on IMC systems. The applicability of A-shift on IMC with other number representations are described in Appendix B.

**Impact of W-reshape and A-shift** Fig. 3b summarizes the results obtained by applying W-reshape and A-shift on ImageNet dataset. A particularly useful view is looking at the distribution of the ADC input. We consider the utilization of ADC quantization range to quantitatively analyze the results, i.e., $\frac{\text{\# of occupied ADC quantization intervals}}{\text{Total ADC quantization invervals}}$. Fig. 3b (top) shows an example of an 8-bit ADC, resulting in 3.52% and 21.7% utilization without and with W-reshape and A-shift, respectively. We also compute the variance of these two cases to justify our results, which leads to 0.094 and 8.535 respectively. These improvements can be directly related to the increased SQNR illustrated in Fig. 3b (bottom).

## 4.2 SQNR ADAPTATION FOR NEURAL NETWORKS

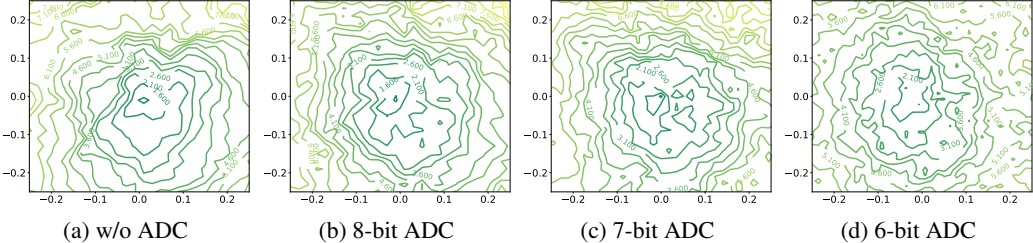

| (a) w/o ADC | (b) 8-bit ADC | (c) 7-bit ADC | (d) 6-bit ADC |

Figure 4: Loss surfaces with A-shift and W-reshape applied for 4-bit activations and weights.

**Bit augmentation (BitAug).** Quantization fundamentally sacrifices information in exchange for model compression. While the SQNR is improved through Eq. 4 - Eq. 7, quantization imposed by the ADC is observed to make SGD-based optimization more challenging during training. Fig. 4 shows the loss surfaces of MobileNetV2 in two randomly selected parameter dimensions for visualization (Li et al., 2018). As seen, ADC quantization causes a less smooth surface with additional local minima. These attributes reduce the likelihood of arriving at preferred (low-loss) minima during the training process. Approaches are thus required to adapt the model to this extra quantization. We seek an approach that facilitates a greater volume of information to be backpropagated so that the model parameters can be optimized more effectively. Inspired by NetAug (Cai et al., 2021) where a tiny model is inserted into larger models during training, we augment the network with ADCs of different bit precisions. At each iteration, we first pass the desired ADC bit to the model and then pass other bit precisions from a pre-defined set $\mathbb{B}$ to the model. The general form of BitAug is

$$\mathcal{L}_A = \mathcal{L}(\theta, b_a) + \lambda_b \sum_{i=1}^{B} \mathcal{L}(\theta, b_{a,i}), \tag{8}$$

where $\theta$ denotes the network parameters, $\lambda_b$ is the coefficient of the BitAug loss, $B$ is the size of the BitAug set $\mathbb{B}$, and $b_{a,i}$ is a sample from the set. Elements in $\mathbb{B}$ are chosen to be neighbors of the target ADC bit precision. Given the complexity of optimization with ADC quantization, we simply employ the assistance of other bit precisions. The information associated with the various ADC bit

precisions is subsequently represented in their respective gradients, which get accumulated during the backward path for more optimal updating of model parameters, i.e.,

$$\theta^{t+1} = \theta^t - \eta \frac{\partial \mathcal{L}(\theta^t, b_a)}{\partial \theta^t} - \eta \lambda_b \sum_{i=1}^{B} \frac{\partial \mathcal{L}(\theta^t, b_{a,i})}{\partial \theta^t}, \tag{9}$$

where $t$ indicates the current training step and $\eta$ denotes learning rate. However, such an aggregation of multiple augmented models is computationally expensive. Following a similar strategy as (Cai et al., 2021), we randomly sample an ADC bit precision for each iteration, i.e.,

$$\mathcal{L}_A = \mathcal{L}(\theta, b_a) + \lambda_b \mathcal{L}(\theta, \widetilde{b}_a), \tag{10}$$

where $\widetilde{b}_a$ is a uniformly sampled bit precision from $\mathbb{B}$. We observe that doing this not only improves the computational efficiency by a factor of $B$, but also achieves better performance than running all ADC bit precisions simultaneously. We include a quantitative study in Appendix C. The selection of $\mathbb{B}$ is also critical. For instance, if we only sample lower precision ADC, we are essentially adding noise to the training process, which causes accuracy degradation. Our empirical results show that a good choice is to choose 1-bit lower and 2-bit higher than the desired ADC bit precision, i.e., $\mathbb{B} = \{b_a - 1, b_a + 1, b_a + 2\}$. A more detailed analysis of BitAug is provided in Appendix F.

## 5 EXPERIMENTS

### 5.1 EXPERIMENTAL SETUP

We consider a general IMC architecture as shown in Fig. 1a. While exploring different IMC architectures is not the focus of this paper, we include experiments on the impact of RAOQ on various IMC configurations in Appendix D for interested readers. In this section, we focus on an IMC system with aggressive memory dimensions of $512 \times 512$, taking 4-bit inputs and processing 4-bit weights, with ADCs having $k = 4$. Higher-precision activations and weights are mapped to the IMC via matrix tiling.

The proposed methods are evaluated on different AI tasks. To preserve the fidelity of critical information, we do not map depthwise convolutions in the MobileNet family, and the second matrix-matrix multiplication in the self-attention module of BERT (BMM2) to the IMC system. This is justified as these layers account for a small number of computations in the overall model (i.e., $< 7\%$ for depthwise convolutions in MobileNetV2 and $< 1.5\%$ for BMM2 in BERT), thus giving minor energy-efficiency advantage by execution via IMC. The first and last layers are kept in 8-bit. We start from pre-trained FP32 models, and first perform QAT based on LSQ+ (Bhalgat et al., 2020) on activations and weights with the proposed W-reshape and A-shift methods. We then add ADC quantization along with other RAOQ techniques for another stage of training. Experiments are performed on Nvidia A100 GPUs. Further training details are described in Appendix E.

### 5.2 RESULTS

Table 1 summarizes the results for 4-bit and 8-bit activations and weights. We sweep the ADC bit precision to demonstrate the robustness and generalizability of our approaches. All QAT (without ADC involved) accuracy matches SOTA results. For a fair comparison, we also perform conventional QAT (i.e., without any proposed methods involved) for ADC quantization. As seen, the proposed RAOQ significantly outperforms conventional QAT in all cases.

**Image classification.** We choose ResNet, MobileNet, and EfficientNet-lite models for evaluation using top-1 accuracy on ImageNet dataset. Our proposed RAOQ restores the performance to high accuracy across the activation/weight and ADC bit precisions considered. Particularly, some cases of 9-bit ADC even outperform the no-ADC baseline. We start to observe accuracy degradation at low precision ADCs, with $\leq 0.3\%$ drop in the 8-bit case, and with $< 0.8\%$ drop in the 7-bit case.

**Object detection.** We evaluate YOLOv5s on COCO 2017 dataset in terms of mAP. Although YOLOv5s involves more complicated network structures compared to the above CNN models for image classification, our approach restores the significantly degraded accuracy to the level close to no-ADC case, with a $< 1\%$ drop for 8-bit and 9-bit ADCs, and $< 2\%$ drop for 7-bit ADC.

Table 1: Evaluation of RAOQ with various activation, weight, and ADC bit precisions.

| Model | FP32 | $b_x, b_w$ | No ADC | $b_a = 7$ | | $b_a = 8$ | | $b_a = 9$ | |
|---|---|---|---|---|---|---|---|---|---|
| | | | | QAT* | RAOQ | QAT* | RAOQ | QAT* | RAOQ |
| ResNet18 | 69.76 | 8,8 | 70.66 | 60.12 | 70.28 | 66.03 | 70.46 | 66.65 | 70.60 |
| | | 4,4 | 70.49 | 59.42 | 70.23 | 65.71 | 70.45 | 66.61 | 70.49 |
| ResNet50 | 76.23 | 8,8 | 76.53 | 65.47 | 76.25 | 73.83 | 76.46 | 75.01 | 76.51 |
| | | 4,4 | 76.31 | 65.25 | 76.15 | 72.05 | 76.27 | 74.16 | 76.32 |
| MobileNetV2 | 71.81 | 8,8 | 71.89 | 62.09 | 71.57 | 66.72 | 71.79 | 69.13 | 71.93 |
| | | 4,4 | 70.47 | 61.51 | 70.22 | 66.67 | 70.46 | 68.55 | 70.45 |
| EfficientNet-lite0 | 75.12 | 8,8 | 74.31 | 61.27 | 73.58 | 68.11 | 74.08 | 68.85 | 74.21 |
| | | 4,4 | 72.84 | 61.21 | 72.18 | 67.03 | 72.76 | 67.85 | 72.82 |
| YOLOv5s | 37.20◇ | 8,8 | 36.60 | 1.30 | 34.73 | 8.02 | 35.82 | 24.03 | 36.41 |
| | | 4,4 | 33.78 | 10.13 | 32.23 | 20.32 | 33.49 | 28.49 | 33.89 |
| BERT-base | 88.58 | 8,8 | 88.24 | 66.35 | 87.40 | 83.04 | 87.84 | 84.82 | 88.11 |
| | | 4,4 | 87.75 | 64.46 | 87.31 | 82.43 | 87.67 | 84.53 | 87.75 |
| BERT-large | 91.00 | 8,8 | 90.58 | 58.37 | 89.60 | 79.58 | 90.09 | 85.92 | 90.38 |
| | | 4,4 | 89.57 | 62.11 | 88.67 | 80.18 | 89.08 | 85.01 | 89.55 |

* Conventional QAT (i.e., without RAOQ techniques) with ADC quantization present. ◇ Result trained by ourselves in FP32 rather than original mixed-precision.

**NLP**. We use BERT models, implemented based on (Wolf et al., 2020), to demonstrate for the question-answering task on SQuAD 1.1 dataset. The results are evaluated in terms of the F1 score. Once again, our proposed RAOQ successfully restores the degraded accuracy, with $< 1\%$, $< 0.5\%$, and $< 0.2\%$ accuracy drops for 7-bit, 8-bit, and 9-bit ADCs, respectively.

## 5.3 COMPARISON WITH OTHER METHODS

As mentioned, previous algorithmic works focus on ADC quantization in IMC on small datasets. Thus, Table 2 shows a comparison of our proposed RAOQ approach with other works on the CIFAR-10 dataset. These works are based on various memory technologies (e.g., SRAM, ReRAM). For a fair comparison, we construct the same model, following the same configurations as these works (e.g., bit precisions, memory dimensions, applicable hardware noise levels), and then apply our RAOQ approach. We see that RAOQ outperforms all other methods, leading to much less degradation regardless of IMC technology and configurations.

Table 2: Comparison of different methods for ADC quantization on CIFAR-10. $M$ denotes the memory inner-dimension, and the column IMC indicates accuracy under ADC quantization.

| Model | Method | $b_x, b_w, b_a$ | $M$ | FP32 | No ADC | IMC | Degradation |
|---|---|---|---|---|---|---|---|
| ResNet20 | (Jin et al., 2022) | 4,4,7 | 9 | – | 91.60 | 91.00 | -0.60[b] |
| | | 4,4,3 | 9 | | | 81.70 | -9.30[b] |
| | RAOQ | 4,4,7 | 9 | 92.32 | 92.23 | 92.32 | +0.09[b] |
| | | 4,4,3 | 9 | | | 89.34 | -2.89[b] |
| ResNet18[a] | (Sun et al., 2021) | 4,4,4 | 256 | 88.87 | – | 86.55 | -2.32[c] |
| | RAOQ | 4,4,4 | 256 | 92.10 | 92.13 | 90.48 | -1.65[c] |
| ResNet18 | (Wei et al., 2020) | 2,2,4 | 9 | 92.01 | 89.62 | 83.37 | -6.25[b] |
| | | 2,2,4 | 36 | | | 87.56 | -2.06[b] |
| | RAOQ | 2,2,4 | 9 | 93.21 | 92.26 | 91.90 | -0.36[b] |
| | | 2,2,4 | 36 | | | 91.81 | -0.45[b] |

[a] Channels are reduced to 1/4 of the original ResNet18. [b] Accuracy drop of IMC ADC quantization with respect to no-ADC case. [c] Accuracy drop with respect to FP32.

## 5.4 ABLATION STUDY

We investigate the impact of each proposed technique in RAOQ. In particular, we use BERT-base, MobileNetV2, and ResNet50 with 4-bit activations and weights, and an 8-bit ADC to perform the study. The results are summarized in Table 3. The first row corresponds to the case where conventional QAT methods are applied to IMC with ADC quantization. Each check mark indicates the presence of a specific technique. As seen, all of the proposed techniques improve the degraded performance due to ADC quantization. Comparatively, A-shift and BitAug exhibit more significant impacts on the network performance, one contributing to boosting SQNR and the other responsible for model optimization.

Table 3: Impact of different methods. The check mark indicates the use of the corresponding method.

| A-shift | W-reshape | BitAug | BERT-base | MobileNetV2 | ResNet50 |
|---|---|---|---|---|---|
| | | | 82.43 | 66.67 | 72.05 |
| $\checkmark$ | | | 84.24 | 68.07 | 75.77 |
| | $\checkmark$ | | 83.06 | 67.61 | 75.01 |
| | | $\checkmark$ | 85.10 | 68.13 | 75.65 |
| $\checkmark$ | $\checkmark$ | | 86.12 | 69.73 | 76.02 |
| $\checkmark$ | $\checkmark$ | $\checkmark$ | 87.67 | 70.46 | 76.27 |

## 6 IMC SYSTEM PERFORMANCE

In this section, we analyze the value of our proposed approaches in handling ADC quantization. Generally, scaling up ADC bit precision brings costs in hardware energy, as shown in Fig. 5 based on a survey of design reported in the literature (Murmann). The ADC energy cost scales considerably at higher precision, and thus directly affects the energy efficiency advantages of IMC systems. Fig. 5 further depicts the IMC energy efficiency for various ADC precisions, compared to fully-optimized digital accelerators. The IMC efficiency is modeled from (Lee et al., 2021a), while the digital-accelerator energy is from (Jouppi et al., 2017), both in the same silicon technology (28nm CMOS), measured as the number of Tera operations per second per Watt (TOPS/W) for 8-bit activation and weight computations. While IMC demonstrates a dramatic energy efficiency advantage over digital accelerators, the advantage drops significantly as ADC precision is increased. With the observed trade-off between conventional QAT-based inference accuracy and energy efficiency, our proposed algorithmic RAOQ approach enables significant improvement in this trade-off.

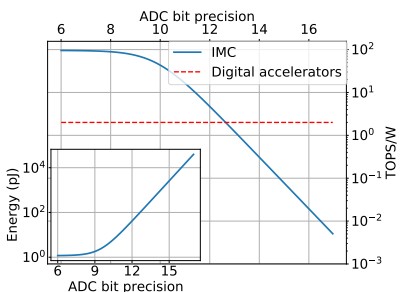

Figure 5: Energy efficiency of IMC.

## 7 CONCLUSION

Analog IMC has shown substantial promise to simultaneously enhance compute efficiency and data-movement costs for AI inference. However, the associated ADC quantization restricts the accuracy of SOTA models applied to challenging tasks. While increasing ADC bit precision reduces the effects of quantization, this comes with a significant energy cost. In this work, we propose RAOQ to tackle such quantization. Specifically, we propose W-reshape and A-shift, to maximize the SQNR following ADC quantization via adjusting the statistics of weights and activations. We further propose BitAug to improve the optimization process. Our work has been evaluated on various datasets, models, and bit precisions, achieving consistently high accuracy. The generalizability and robustness of our proposed methods demonstrate the feasibility of applying IMC to challenging AI tasks.

## 8 REPRODUCIBILITY

The detailed training configurations are described in Appendix E, including the training procedure, hyperparameter settings, learning curves, as well as compute resources needed to perform our experiments. The training of each model is described separately for clarity. In Appendix E.4, we provide a code example to implement our proposed RAOQ method, associated with a sample log file of MobileNetV2 training. For readers who are interested in other IMC configurations, we provide studies on different IMC configurations in Appendix D, other than those presented in the main manuscript. All of our proposed methods can be directly applied except that a small adjustment needs to be made for A-shift for different IMC types, as detailed in Appendix B.

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

## A    IMPACT OF W-RESHAPE ON INFERENCE ACCURACY

In Section 4.1 of the paper, we introduce a weight reshaping method (W-reshape) to adjust weight statistics to improve SQNR following ADC quantization. In this section, we study the impact of W-reshape on inference accuracy. Specifically, we use ResNet50 as an example in this study. We first visualize kurtosis with different $\lambda_\kappa$ in Fig. 6a by computing the kurtosis for quantized weights at each layer. As seen, increasing $\lambda_\kappa$ reduces kurtosis, but saturates when $\lambda_\kappa$ becomes too large. We also observe that weights in the later layers are more resistive to the effects of kurtosis loss. As shown in Fig. 6b, the blue dots represent the case when we apply a constant $\lambda_\kappa$ to all layers, where we can observe larger kurtosis in later layers. We can therefore adjust the kurtosis-loss coefficient for these layers, applying $4\times$ higher weighting, i.e.,

$$\mathcal{L}_Q = \mathcal{L}_c + \lambda_\kappa \left( \sum\nolimits_{l=1}^{J} \kappa_l + 4\sum\nolimits_{l=J+1}^{L} \kappa_l \right) \tag{11}$$

where $L$ is the number of layers, and $J$ is the boundary to split front layers and later layers. The result is illustrated as orange dots in Fig. 6b, which show reduced kurtosis in later layers.

Table 4 and Table 5 show both the ResNet50 and MobileNetv2 accuracy of QAT (i.e., without ADC) and the accuracy after incorporating ADC quantization under different strengths of the kurtosis loss. As we can see, there are clearly a trade-off between QAT accuracy and the amount of kurtosis loss applied, which therefore impacts the overall accuracy with ADC quantization. First, we can see that small $\lambda_\kappa$ provides slightly higher accuracy for QAT without ADC quantization. However, large kurtosis of the quantized-weight distribution leads to low variance of the IMC compute output (ADC input). Consequently, accuracy after incorporating ADC quantization is low. An extremely large $\lambda_\kappa$ starts to degrade accuracy of QAT without ADC quantization, and thus limits the accuracy achievable after incorporating ADC quantization, despite larger variance of the IMC compute output. This can be further understood by plotting the distributions of quantized weights for each $\lambda_\kappa$, as shown in Fig. 6c-6e. As seen, a large $\lambda_\kappa$ leads to significant clipping error, eliminating almost all information, and thus resulting in degraded accuracy. Therefore, in this work, we choose $\lambda_\kappa = 0.0005$ to maximize the variance of quantized weights with accuracy degradation less than $0.1\%$ during the QAT stage (i.e., without ADC quantization), but boosting $> 0.3\%$ accuracy when incorporating ADC quantization.

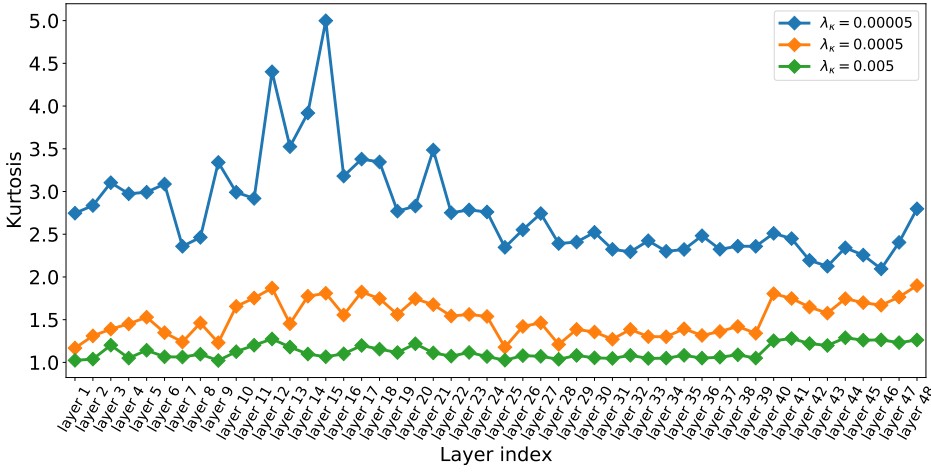

(a) Kurtosis of each layer.

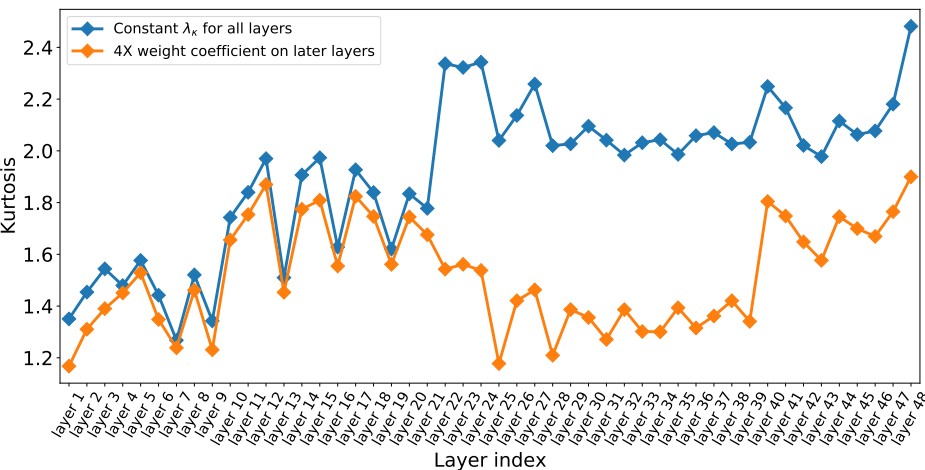

(b) Comparison of different strategies to assign $\lambda_\kappa$ to neural network layers.

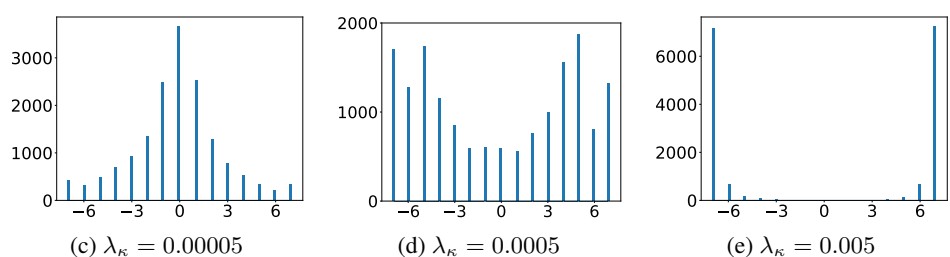

(c) $\lambda_\kappa = 0.00005$        (d) $\lambda_\kappa = 0.0005$        (e) $\lambda_\kappa = 0.005$

Figure 6: Visualize the impact of W-reshape on quantized weights.

Table 4: ResNet accuracy with different $\lambda_\kappa$.

| $\lambda_\kappa$ | 0 | 0.000025 | 0.00005 | 0.0005 | 0.005 | 0.01 |
|---|---|---|---|---|---|---|
| Accuracy (w/o ADC) | 76.35 | 76.36 | 76.32 | 76.31 | 76.15 | 75.51 |
| Accuracy (w/ ADC) | 75.91 | 75.92 | 76.05 | 76.27 | 75.77 | 75.02 |

Table 5: MobileNetv2 accuracy with different $\lambda_\kappa$.

| $\lambda_\kappa$ | 0 | 0.00004 | 0.00065 | 0.0008 | 0.002 | 0.01 |
|---|---|---|---|---|---|---|
| Accuracy (w/o ADC) | 70.44 | 70.51 | 70.47 | 70.33 | 69.98 | 69.05 |
| Accuracy (w/ ADC) | 69.92 | 70.02 | 70.46 | 70.24 | 69.65 | 68.86 |

## B  IMC COMPATIBILITY

All techniques in RAOQ are compatible generally across IMC hardware. W-reshape and BitAug simply impact the weight parameters derived from neural network training. A-shift is a little different, in that it is affected by how activations are mapped for IMC computation after training, and here we examine its impact from different IMC hardware approaches. Previous IMC works employ different ways of encoding multi-bit activations and weights. For example, Dong et al. (2020) follows conventional 2's complement format, which we refer to as 0/1 representation, corresponding to the mathematical value of individual binary-weighted bits. However, other works like (Lee et al., 2021a) represent a multi-bit number with individual binary-weighted bits taking mathematical values of -1 or 1, thus enabling multiplication simply by performing logical XNOR operations. We refer to this format as -1/1 representation. These two types of number representations are illustrated in Fig. 7, taking 2-bit as an example. In Section 4.1 of the paper, we show A-shift for 0/1 representation, which is the default number representation in neural network training. In fact, our proposed A-shift can be easily adapted to -1/1 representation as well. This is because these two representations can be converted to each other via a linear transformation. Let $\overline{x}_{0/1}$ and $\overline{x}_{-1/1}$ denote the IMC input for 0/1 representation and -1/1 representation, respectively, then:

$$\overline{x}_{-1/1} = 2\,\overline{x}_{0/1} + 1. \tag{12}$$

Therefore, A-shift for -1/1 representation can be expressed as:

$$\overline{x}_s = 2\left\lfloor clip\left(\frac{x-\beta}{s_x}, 0, 2^{b_x}-1\right)\right\rceil - (2^{b_x}-1) \tag{13}$$

$$= 2\,\overline{x}_{-1/1} - (2^{b_x}-1). \tag{14}$$

We can see that while A-shift for 0/1 representation shifts the range from $\{n : n \in \mathbb{Z} \text{ and } n \geq 0 \text{ and } n \leq 2^{b_x}-1\}$ to $\{n : n \in \mathbb{Z} \text{ and } n \geq -2^{b_x-1} \text{ and } n \leq 2^{b_x-1}-1\}$, A-shift for -1/1 representation shifts to $\{2n+1 : n \in \mathbb{Z} \text{ and } n \geq -2^{b_x-1} \text{ and } n \leq 2^{b_x-1}-1\}$. Similar to the case of 0/1 representation, the extra offset introduced by A-shift can be computed offline. In summary, all of our proposed approaches are compatible with various IMC types.

| Number | Representation |
|---|---|
| -2 | 1 0 |
| -1 | 1 1 |
| 0 | 0 0 |
| 1 | 0 1 |

(a) 0/1 representation.

| Number | Representation |
|---|---|
| -3 | -1 -1 |
| -1 | -1 1 |
| 1 | 1 -1 |
| 3 | 1 1 |

(b) -1/1 representation.

(c) IMC.

Figure 7: (a-b) Number representations of different IMCs. (c) Example IMC system using -1/1 representation.

## C  CHOOSING BIT PRECISION CANDIDATES FOR BITAUG

In Section 4.2 of the paper, we introduced BitAug, and proposed the optimal bit precision candidate set as containing 1-bit lower and 2-bit higher than the target ADC bit precision. In this

Table 6: Accuracy of different choices of candidate sets.

| Model | {} | {−2, −1} | {−1, 1} | {+1, +2} | {−1, 1, 2} | {−2, −1, +1, +2} | {−1, +1, +2, +3} |
|---|---|---|---|---|---|---|---|
| MobileNetv2 | 68.45 | 66.65 | 68.98 | 68.70 | 69.92 | 69.11 | 69.41 |
| ResNet50 | 74.47 | 71.12 | 73.37 | 73.42 | 75.83 | 74.77 | 75.21 |
| BERT-base | 85.45 | 82.13 | 85.41 | 86.88 | 86.92 | 86.04 | 86.58 |

section, we first explore the impact of different candidate sets. Table 6 shows the model accuracy of different candidates using MobileNetV2, ResNet50, and BERT-base as examples, with 4-bit activations/weights and 8-bit ADCs. In order to see the more obvious effects from BitAug, we use ADCs with $k = 1$, i.e., more quantization noise. Each candidate set is represented by the offset of element values from the target ADC bit precision $b_a$, e.g., $\{−2, −1\}$ implies $\{b_a − 2, b_a − 1\}$. As seen, our proposed candidate set achieves the highest accuracy.

We also explore the relationship between the training complexity and the model accuracy. Table 7 compares the accuracy achieved by randomly sampling one element from the candidate set as well as running all elements from the candidate set simultaneously. As seen, our proposed execution of BitAug not only has lower computational complexity compared to running all bits at once, but demonstrates higher accuracy.

Table 7: Evaluation accuracy by executing BitAug in different modes.

| Mode | Sample single bit | Run all candidates |
|---|---|---|
| Accuracy | 69.92 | 69.04 |

## D  IMPACT OF IMC CONFIGURATIONS

In this section, we explore our proposed RAOQ under different hardware configurations. First, we choose ResNet50, MobileNetV2, and BERT-base for different IMC rows, corresponding to the inner-product accumulation dimension. Table 8 shows the evaluation accuracy for different memory inner-product dimensions with 4-bit activations and weights, and 8-bit ADCs having $k = 4$, demonstrating the consistently high accuracy. As we can see, our proposed RAOQ is robust across different memory sizes, indicating promise for deriving substantial benefits from in-memory computing. We can also see that different models slightly favor different memory dimensions. For example, ResNet50 and MobileNetV2 degrade slightly for the case of 256 rows, while BERT-base achieves the best accuracy in this case. This is related to the size of different neural network layers and their mapping to IMC systems, which is out of the scope of this work.

Table 8: Evaluation accuracy of different memory inner-dimensions.

| # of rows | 128 | 256 | 512 | 768 | 1024 |
|---|---|---|---|---|---|
| ResNet50 | 76.33 | 76.28 | 76.27 | 76.31 | 76.24 |
| MobileNetV2 | 70.53 | 70.40 | 70.46 | 70.45 | 70.43 |
| BERT-base | 87.45 | 87.81 | 87.67 | 87.59 | 87.32 |

Fig. 9 shows the performance of our proposed RAOQ with different values of $k$, i.e., the clipping set by hardware designers. As observed, these models generally favor some clipping in exchange for finer ADC quantization steps. Despite employing aggressive clipping with $k = 8$, these models preserve relatively high accuracy. This can be attributed to the fact that even with the use of our proposed W-reshape and A-shift, the distribution of ADC-input data still concentrates within a narrow portion of the entire ADC range. Therefore, ADC quantization error still dominates clipping error. However, a significant degradation in model accuracy becomes apparent when $k$ is set to 16, even with the help of our proposed RAOQ. At this point, clipping errors start to dominant, leading to considerable loss of information.

Table 9: Evaluation accuracy of different $k$.

| $k$ | 1 | 2 | 4 | 8 | 16 |
|---|---|---|---|---|---|
| ResNet50 | 75.84 | 76.21 | 76.27 | 76.13 | 75.67 |
| MobileNetV2 | 69.92 | 70.36 | 70.46 | 70.48 | 69.47 |
| BERT-base | 86.74 | 86.97 | 87.67 | 87.28 | 6.62 |

## E   TRAINING DETAILS

In this section, we describe the models, datasets, and hyperparameter settings used in our experiments. We implement our models in the PyTorch framework. The first and last layers are kept in 8-bit, and are not mapped to the IMC. Mapping these layers to IMC provides marginal benefit, since the first layers have few input channels and thus limited opportunity for row parallelism, while the last layers have low data reuse, contributing a small number of total operations and restricting amortization of the IMC weight-loading overheads. $\lambda_b$ for BitAug is initialized to be 1, and drops following a cosine scheduling for all models. We first perform QAT for these models based on LSQ+ (Bhalgat et al., 2020) without ADC quantization. Then we perform our proposed RAOQ with ADC quantization incorporated. The training overhead of A-shift and W-reshape is negligible. BitAug increases the GPU memory usage by $14\%$ and reduces the training speed by $1.5\times$ during the last training phase, due to the accumulation of gradients associated with different ADC bit precisions. In the following sections, we show the training curves for both QAT stage (without ADC quantization) and the training stage with ADC quantization involved for each model, with 4-bit activations and weights, and 8-bit ADCs as examples.

### E.1   IMAGE CLASSIFICATION

We perform image classification using the ImageNet dataset (Deng et al., 2009), including 1000 classes of objects with over 1.2 million training images and 50,000 validation images. Our experiments consist of models from the ResNet family and the MobileNet family, whose training settings are discussed in detail as follows.

**ResNet18.** For 4-bit QAT (i.e., 4-bit activations and weights), we perform training for 90 epochs, with a batch size of 256. We use SGD optimizer with a momentum of 0.9 and weight decay of 0.0001. The learning rate starts at 0.01 and gradually drops, following a cosine annealing scheduler. $\lambda_\kappa$ is set to 0.002. For 8-bit QAT, we follow the same optimizer and batch size as the 4-bit case. We train for 30 epochs with an initial learning rate of 0.005. $\lambda_\kappa$ is set to 0.0014. When ADC quantization is added to the model, we perform another 30-epoch training, using the cosine annealing learning rate scheduler with an initial learning rate of 0.004. The optimizer and batch size remain the same as those used in the previous QAT phase. All experiments for ResNet18 are conducted on 2 Nvidia A100 GPUs.

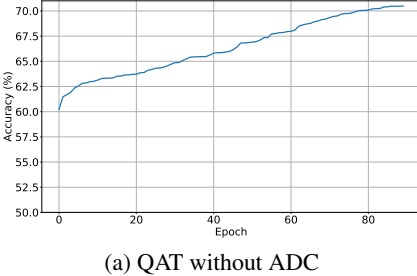
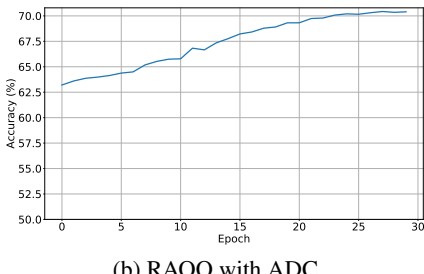

(a) QAT without ADC                    (b) RAOQ with ADC

Figure 8: Training curves for ResNet18.

**ResNet50.** During the QAT phase, we train for 80 epochs, with batch size 256 for 4-bit activations and weights. We use the same optimizer and learning rate scheduler as used for ResNet18. $\lambda_\kappa$ for ResNet50 is set to 0.0005. For 8-bit QAT, we maintain the same optimizer and learning rate

scheduler, but with a different initial learning rate of 0.002. We train for 40 epochs with a batch size of 256. $\lambda_\kappa$ is set to 0.00011. For the next stage incorporating ADC quantization, we train for another 40 epochs with an initial learning rate of 0.002. The rest of the settings are the same as those used for ResNet18. All experiments for ResNet50 are performed on 2 Nvidia A100 GPUs.

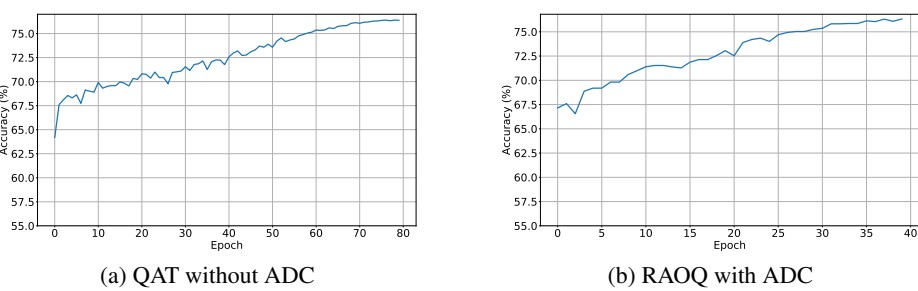

(a) QAT without ADC          (b) RAOQ with ADC

Figure 9: Training curves for ResNet50.

**MobileNetV2.** We perform 4-bit QAT for 70 epochs with batch size 256. We use SGD optimizer with a momentum of 0.9 and weight decay of 0.00004. The initial learning rate is set to 0.01, with a cosine annealing scheduler. $\lambda_\kappa$ is set to 0.00065. For 8-bit QAT, we keep the same optimizer, scheduler, $\lambda_\kappa$, and batch size, training for 40 epochs with an initial learning rate of 0.002. We then perform another phase of training with ADC quantization added for 50 epochs. We use the same optimizer, scheduler, and batch size as used in QAT. The learning rate starts at 0.004. The entire training for MobileNetV2 is on 2 Nvidia A100 GPUs.

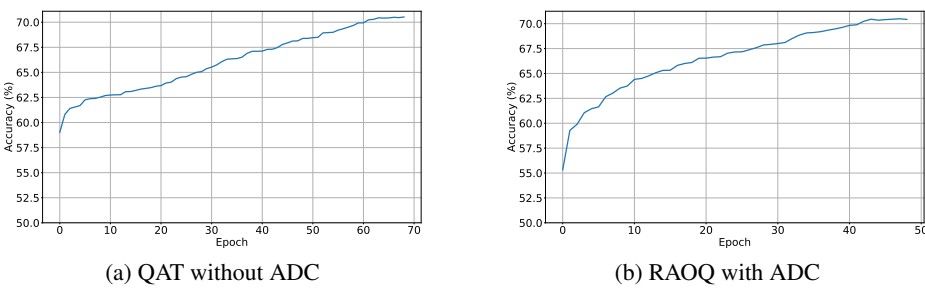

(a) QAT without ADC          (b) RAOQ with ADC

Figure 10: Training curves for MobileNetV2.

**EfficientNet-lite0.** The settings are the same as MobileNetV2, except that $\lambda_\kappa$ is set to 0.002. We perform training for 80 epochs with an initial learning rate of 0.01 for both 4-bit and 8-bit QAT. When ADC quantization is added for the subsequent training phase, we use the same optimizer, scheduler, and batch size, running for 50 epochs. The initial learning rate is set to 0.004. Once again, we perform the experiments on 2 Nvidia A100 GPUs.

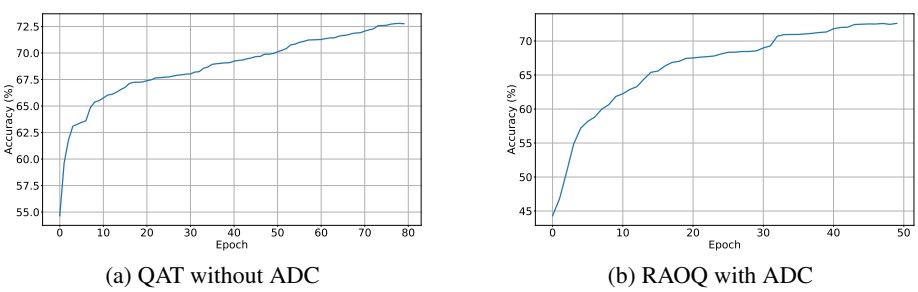

(a) QAT without ADC          (b) RAOQ with ADC

Figure 11: Training curves for EfficientNet-lite0.

### E.2    OBJECT DETECTION

We perform object detection on the COCO dataset (Lin et al., 2014), including 118K images for training and 5K images for validation. YOLOv5s is used to conduct the task.

**YOLOv5s.**    We first retrain the floating point (FP) model by using hyperparameters indicated in (Jocher et al., 2022). However, we remove the automatic mixed precision (AMP) features supported by PyTorch, since we would like to perform quantization to the target bit precisions later on. Most of our hyperparameters remain the same as (Jocher et al., 2022) suggested, and we specify those we customized for this work here. For 4-bit QAT, we train for 100 epochs with a batch size of 64 with a weight decay of 0.0001. The initial learning rate is set to 0.004 following a cosine scheduler, and the momentum for the optimizer is changed to 0.9 from 0.937 in (Jocher et al., 2022). The 8-bit QAT runs for 80 epochs with the same optimizer, scheduler, and batch size as used for the 4-bit QAT, and an adjusted initial learning rate of 0.004. $\lambda_\kappa$ is set to 0.0001. For training with ADC quantization incorporated, we perform the training for 20 epochs for the 4-bit case, with the initial learning rate of 0.0001 and weight decay of 0.00005. We train the 8-bit YOLO model with ADC quantization for 90 epochs, using the same hyperparameters as used in the QAT phase except for the batch size increased to 128. Our experiments are performed on 4 Nvidia A100 GPUs.

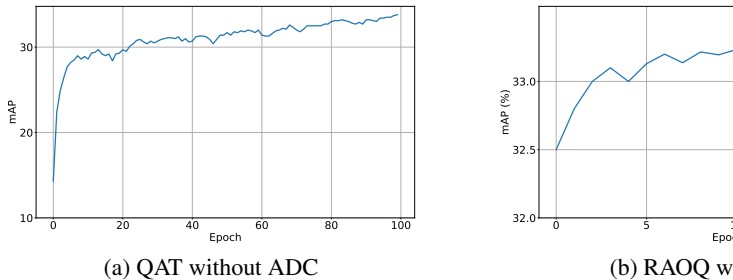

(a) QAT without ADC    (b) RAOQ with ADC

Figure 12: Training curves for YOLOv5s.

### E.3    NATURAL LANGUAGE PROCESSING (NLP)

We perform the question-answering task on SQuAD 1.1 (Rajpurkar et al., 2016), a reading comprehension dataset, containing more than 100k question-and-answer pairs. We use BERT-base and BERT-large to accomplish this task. As only a small number of activations in the BERT models follow non-linear functions (i.e., GELU), we can apply A-shift to only these layers. Activations from other layers are directly taken as the result of matrix multiplications. We find that forcing their quantization to unsigned numbers causes accuracy degradation. The sequence length is kept at 384 for all training stages and all models. BMM2 layers in BERT are kept in 8-bit for fidelity reasons. As suggested by (Wang et al., 2022), we first fine-tune the pre-trained FP BERT-base for 2 epochs to adapt to the downstream SQuAD 1.1 dataset before any quantization gets involved. This is performed on a single Nvidia A100 GPU with an initial learning rate of 0.00003 and a batch size of 12.

**BERT-base.** For this model, we use the same hyperparameters for 4-bit and 8-bit QAT. Specifically, we use AdamW as the optimizer and a batch size of 16, running for 4 epochs. The initial learning rate is kept at 0.00003, following a linear decay. The dropout rate is raised to 0.2. The training phase with ADC quantization incorporated uses the same hyperparameters as used in the QAT phase. Experiments for BERT-base with quantization are performed on 2 Nvidia A100 GPUs.

**BERT-large.** For 4-bit QAT, our optimizer, batch size, and learning rate scheduler is the same as those used in BERT-base. We perform training for 8 epochs. We observe that 4-bit QAT is sensitive to the change in dropout rate. Thus, we start with a dropout rate of 0.1 for the first epoch, and then raise to 0.2 for the rest of the training. 8-bit QAT is performed for 4 epochs with a constant dropout rate of 0.2. The rest of the hyperparameters are the same as 4-bit QAT. Regarding training with ADC quantization, we use the same parameters as those used in BERT-base. Experiments in 8-bit and 4-bit are performed on 4 and 2 Nvidia A100 GPUs, respectively.

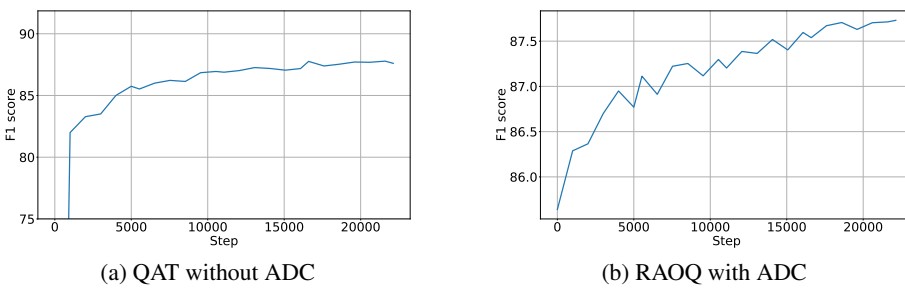

(a) QAT without ADC                    (b) RAOQ with ADC

Figure 13: Training curves for BERT-base.

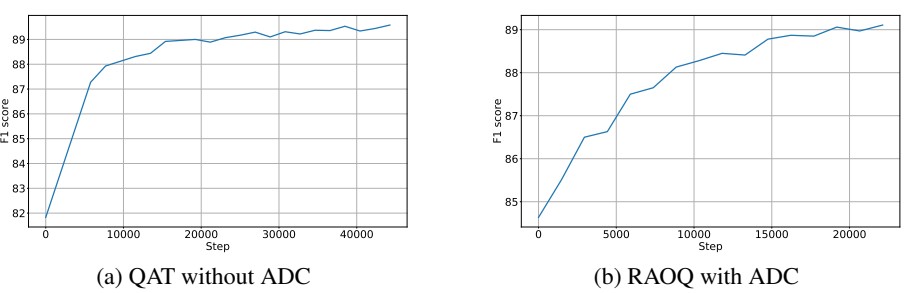

(a) QAT without ADC                    (b) RAOQ with ADC

Figure 14: Training curves for BERT-large.

### E.4 CODE EXAMPLE

All models are implemented using PyToch framework. Specifically, we customized nn.Conv2d and nn.Linear modules to incorporate quantization of activations/weights/ADCs, mapping to IMC systems, and our proposed A-shift and W-reshape methods. The proposed BitAug technique is integrated with top-level training. Fig. 15a-15b shows example code snippets for a convolution layer with all quantization sources integrated and the implementation of our proposed approaches. Fig. 15c illustrates a screenshot of the training log of MobileNetV2.

## F  ANALYSIS OF BITAUG

In this section, we provide further study of BitAug. We argue that BitAug can assist the training process to avoid getting stuck in local minima. To demonstrate this, we consider a toy example, i.e., a single layer neural network with randomly generated input $X \in \mathbb{R}^{M \times M}$ and a randomly generated weight $W \in \mathbb{R}^{M \times N}$, followed by a simple spiking function $g(x) = x^2 - 60cos(x)$ (with global minimum at $x = 0$), as shown in Fig. 16a. We define the loss function as:

$$\mathcal{L} = \sum_{i,j} g(XW)_{i,j} \tag{15}$$

To study the effects of BitAug, we introduce quantizers for the activations, weights, and outputs of this network. Clearly, $\mathcal{L}$ gets its global minimum when all weights are zero. We first perform training without BitAug. The statistics of the weight parameters from the last iteration are shown in Fig. 16b with a standard deviation of $18.07$. Rather than converging to zero, most of the weights are concentrated at some negative value, which indicates a possible local minima. Comparatively, the results after applying BitAug are shown in Fig. 16c with a standard deviation of $12.39$. As seen, it shows a stronger convergence towards zero compared to without BitAug, improving the ability to escape from the local minima.

We further plot the loss surfaces at the pre-trained checkpoints for both cases (i.e., without and with BitAug), illustrated in Fig. 16d and Fig. 16e. We observe a reduction of the number local minimum after applying BitAug. Fig. 16f further confirms this observation, which shows the loss surface with

```python
# quantize activation
xq, scale_x, bias_x = x_quantizer(input, bx)

# quantize weight
wq, scale_w = w_quantizer(self.weight, bw)

# W-reshape
if w_reshape:
    loss_w = compute_w_reshape(wq)

# A-shift
xq_s, bias_s = compute_a_shift(xq, bx)

# IMC
y_analog = imc(xq_s, wq, imc_configs)

# ADC quantization
y = adc_quant(y_analog, ba, imc_configs)

# reconstruct
out_quant = de_a_shift(y, bias_s)

# de-quantize
out = dequant(out_quant, scale_x, scale_w, bias_x)
```

(a)

```python
# forward
output, loss_w = model(data, adc_bit)

# compute loss
loss = compute_loss(output, target)

# apply W-reshape
if loss_w is not None:
    loss += lambda_w * loss_w

# backprop
loss.backward()

# BitAug
# sample ADC bit from candidates
aug_bit = sample_bit(bit_candidates)

# pass sampled ADC bit to NN
output, _ = model(data, aug_bit)
loss_aug = lambda_b * compute_loss(output, target)
loss_aug.backward()

# update
optimizer.step()
```

(b)

```
Epoch: [49][2750/5005]  Time  0.304 ( 0.465)   Data  0.000 ( 0.140)   Loss 1.30e+00 (1.20e+00)   Acc@1  70.31 ( 71.42)   Acc@5  87.50 ( 88.64)
Epoch: [49][2800/5005]  Time  0.297 ( 0.466)   Data  0.000 ( 0.138)   Loss 9.78e-01 (1.20e+00)   Acc@1  77.34 ( 71.43)   Acc@5  89.84 ( 88.65)
Epoch: [49][2850/5005]  Time  0.308 ( 0.467)   Data  0.000 ( 0.135)   Loss 1.15e+00 (1.20e+00)   Acc@1  72.66 ( 71.43)   Acc@5  88.28 ( 88.66)
Epoch: [49][2900/5005]  Time  0.297 ( 0.468)   Data  0.000 ( 0.133)   Loss 1.28e+00 (1.20e+00)   Acc@1  64.84 ( 71.43)   Acc@5  87.50 ( 88.66)
Epoch: [49][2950/5005]  Time  0.308 ( 0.469)   Data  0.000 ( 0.131)   Loss 1.36e+00 (1.20e+00)   Acc@1  68.75 ( 71.43)   Acc@5  85.16 ( 88.66)
Epoch: [49][3000/5005]  Time  0.297 ( 0.470)   Data  0.000 ( 0.129)   Loss 1.24e+00 (1.20e+00)   Acc@1  66.41 ( 71.42)   Acc@5  86.72 ( 88.65)
Epoch: [49][3050/5005]  Time  0.307 ( 0.470)   Data  0.000 ( 0.127)   Loss 1.13e+00 (1.20e+00)   Acc@1  71.09 ( 71.41)   Acc@5  90.62 ( 88.64)
Epoch: [49][3100/5005]  Time  0.297 ( 0.471)   Data  0.000 ( 0.125)   Loss 1.00e+00 (1.20e+00)   Acc@1  77.34 ( 71.41)   Acc@5  90.62 ( 88.64)
Epoch: [49][3150/5005]  Time  0.323 ( 0.472)   Data  0.044 ( 0.123)   Loss 7.65e-01 (1.20e+00)   Acc@1  82.03 ( 71.41)   Acc@5  94.53 ( 88.64)
Epoch: [49][3200/5005]  Time  0.297 ( 0.473)   Data  0.000 ( 0.122)   Loss 1.21e+00 (1.20e+00)   Acc@1  68.75 ( 71.41)   Acc@5  89.84 ( 88.64)
Epoch: [49][3250/5005]  Time  0.308 ( 0.473)   Data  0.000 ( 0.120)   Loss 1.11e+00 (1.19e+00)   Acc@1  75.78 ( 71.41)   Acc@5  89.06 ( 88.65)
Epoch: [49][3300/5005]  Time  0.297 ( 0.474)   Data  0.000 ( 0.119)   Loss 1.32e+00 (1.20e+00)   Acc@1  70.31 ( 71.41)   Acc@5  89.06 ( 88.65)
Epoch: [49][3350/5005]  Time  0.307 ( 0.475)   Data  0.000 ( 0.117)   Loss 1.05e+00 (1.19e+00)   Acc@1  70.31 ( 71.41)   Acc@5  91.41 ( 88.65)
Epoch: [49][3400/5005]  Time  0.297 ( 0.476)   Data  0.000 ( 0.116)   Loss 1.10e+00 (1.19e+00)   Acc@1  71.88 ( 71.41)   Acc@5  92.19 ( 88.65)
Epoch: [49][3450/5005]  Time  0.307 ( 0.476)   Data  0.000 ( 0.114)   Loss 1.19e+00 (1.19e+00)   Acc@1  75.78 ( 71.40)   Acc@5  90.62 ( 88.64)
Epoch: [49][3500/5005]  Time  0.297 ( 0.477)   Data  0.000 ( 0.113)   Loss 1.22e+00 (1.20e+00)   Acc@1  70.31 ( 71.40)   Acc@5  87.50 ( 88.64)
Epoch: [49][3550/5005]  Time  0.308 ( 0.478)   Data  0.000 ( 0.111)   Loss 1.12e+00 (1.19e+00)   Acc@1  75.00 ( 71.39)   Acc@5  87.50 ( 88.63)
Epoch: [49][3600/5005]  Time  0.297 ( 0.479)   Data  0.000 ( 0.110)   Loss 1.15e+00 (1.19e+00)   Acc@1  70.31 ( 71.39)   Acc@5  90.62 ( 88.63)
Epoch: [49][3650/5005]  Time  0.306 ( 0.479)   Data  0.000 ( 0.108)   Loss 1.32e+00 (1.19e+00)   Acc@1  64.84 ( 71.40)   Acc@5  86.72 ( 88.63)
Epoch: [49][3700/5005]  Time  0.297 ( 0.480)   Data  0.000 ( 0.107)   Loss 1.06e+00 (1.20e+00)   Acc@1  74.22 ( 71.40)   Acc@5  89.06 ( 88.64)
Epoch: [49][3750/5005]  Time  0.307 ( 0.480)   Data  0.000 ( 0.106)   Loss 1.15e+00 (1.20e+00)   Acc@1  71.88 ( 71.39)   Acc@5  89.84 ( 88.62)
Epoch: [49][3800/5005]  Time  0.296 ( 0.481)   Data  0.000 ( 0.105)   Loss 1.25e+00 (1.20e+00)   Acc@1  75.00 ( 71.40)   Acc@5  84.38 ( 88.62)
Epoch: [49][3850/5005]  Time  0.307 ( 0.482)   Data  0.000 ( 0.104)   Loss 8.79e-01 (1.19e+00)   Acc@1  71.88 ( 71.40)   Acc@5  96.09 ( 88.62)
Epoch: [49][3900/5005]  Time  0.297 ( 0.482)   Data  0.000 ( 0.103)   Loss 1.28e+00 (1.19e+00)   Acc@1  69.53 ( 71.41)   Acc@5  87.50 ( 88.62)
Epoch: [49][3950/5005]  Time  0.308 ( 0.483)   Data  0.000 ( 0.101)   Loss 9.40e-01 (1.20e+00)   Acc@1  72.66 ( 71.41)   Acc@5  92.19 ( 88.63)
Epoch: [49][4000/5005]  Time  0.297 ( 0.484)   Data  0.000 ( 0.100)   Loss 9.94e-01 (1.20e+00)   Acc@1  71.88 ( 71.42)   Acc@5  91.41 ( 88.63)
Epoch: [49][4050/5005]  Time  0.307 ( 0.484)   Data  0.000 ( 0.100)   Loss 1.36e+00 (1.20e+00)   Acc@1  69.53 ( 71.43)   Acc@5  86.72 ( 88.63)
Epoch: [49][4100/5005]  Time  0.778 ( 0.485)   Data  0.510 ( 0.101)   Loss 1.31e+00 (1.20e+00)   Acc@1  68.75 ( 71.42)   Acc@5  84.38 ( 88.63)
Epoch: [49][4150/5005]  Time  0.308 ( 0.485)   Data  0.000 ( 0.100)   Loss 1.25e+00 (1.20e+00)   Acc@1  72.66 ( 71.41)   Acc@5  88.28 ( 88.64)
Epoch: [49][4200/5005]  Time  0.297 ( 0.486)   Data  0.000 ( 0.099)   Loss 1.06e+00 (1.20e+00)   Acc@1  75.00 ( 71.41)   Acc@5  89.84 ( 88.64)
Epoch: [49][4250/5005]  Time  0.308 ( 0.487)   Data  0.000 ( 0.098)   Loss 1.03e+00 (1.20e+00)   Acc@1  71.88 ( 71.41)   Acc@5  90.62 ( 88.63)
Epoch: [49][4300/5005]  Time  0.297 ( 0.487)   Data  0.000 ( 0.098)   Loss 1.12e+00 (1.20e+00)   Acc@1  75.00 ( 71.41)   Acc@5  88.28 ( 88.63)
Epoch: [49][4350/5005]  Time  0.308 ( 0.488)   Data  0.000 ( 0.097)   Loss 1.19e+00 (1.20e+00)   Acc@1  74.22 ( 71.42)   Acc@5  90.62 ( 88.63)
Epoch: [49][4400/5005]  Time  0.297 ( 0.489)   Data  0.000 ( 0.096)   Loss 1.29e+00 (1.20e+00)   Acc@1  71.88 ( 71.42)   Acc@5  87.50 ( 88.64)
Epoch: [49][4450/5005]  Time  0.312 ( 0.489)   Data  0.000 ( 0.095)   Loss 1.10e+00 (1.20e+00)   Acc@1  75.00 ( 71.42)   Acc@5  89.84 ( 88.63)
Epoch: [49][4500/5005]  Time  0.297 ( 0.490)   Data  0.000 ( 0.094)   Loss 1.22e+00 (1.20e+00)   Acc@1  73.44 ( 71.42)   Acc@5  89.06 ( 88.64)
Epoch: [49][4550/5005]  Time  0.308 ( 0.490)   Data  0.000 ( 0.093)   Loss 1.12e+00 (1.20e+00)   Acc@1  73.44 ( 71.42)   Acc@5  88.28 ( 88.64)
Epoch: [49][4600/5005]  Time  0.860 ( 0.491)   Data  0.000 ( 0.092)   Loss 1.30e+00 (1.20e+00)   Acc@1  72.66 ( 71.42)   Acc@5  86.72 ( 88.64)
Epoch: [49][4650/5005]  Time  1.171 ( 0.491)   Data  0.903 ( 0.093)   Loss 9.69e-01 (1.20e+00)   Acc@1  75.00 ( 71.43)   Acc@5  93.75 ( 88.64)
Epoch: [49][4700/5005]  Time  0.297 ( 0.492)   Data  0.000 ( 0.094)   Loss 1.22e+00 (1.20e+00)   Acc@1  70.31 ( 71.44)   Acc@5  89.84 ( 88.65)
Epoch: [49][4750/5005]  Time  1.136 ( 0.492)   Data  0.857 ( 0.096)   Loss 8.94e-01 (1.20e+00)   Acc@1  77.34 ( 71.45)   Acc@5  90.62 ( 88.65)
Epoch: [49][4800/5005]  Time  0.297 ( 0.493)   Data  0.000 ( 0.097)   Loss 1.17e+00 (1.20e+00)   Acc@1  68.75 ( 71.43)   Acc@5  87.50 ( 88.65)
Epoch: [49][4850/5005]  Time  1.185 ( 0.493)   Data  0.917 ( 0.099)   Loss 1.16e+00 (1.20e+00)   Acc@1  71.09 ( 71.44)   Acc@5  91.41 ( 88.65)
Epoch: [49][4900/5005]  Time  0.297 ( 0.494)   Data  0.000 ( 0.100)   Loss 9.56e-01 (1.20e+00)   Acc@1  75.00 ( 71.43)   Acc@5  92.97 ( 88.65)
Epoch: [49][4950/5005]  Time  1.299 ( 0.494)   Data  1.031 ( 0.102)   Loss 1.25e+00 (1.20e+00)   Acc@1  69.53 ( 71.43)   Acc@5  89.84 ( 88.65)
Epoch: [49][5000/5005]  Time  0.297 ( 0.495)   Data  0.000 ( 0.102)   Loss 1.09e+00 (1.20e+00)   Acc@1  68.75 ( 71.43)   Acc@5  92.97 ( 88.64)
training time: 2475.8788629059854
Test: [  0/391] Time  3.182 ( 3.182)   Loss 5.4044e-01 (5.4044e-01)   Acc@1  88.28 ( 88.28)   Acc@5  95.31 ( 95.31)
Test: [ 50/391] Time  0.482 ( 0.727)   Loss 3.4430e-01 (9.0779e-01)   Acc@1  89.06 ( 77.39)   Acc@5  99.22 ( 92.62)
Test: [100/391] Time  0.799 ( 0.715)   Loss 8.0320e-01 (9.0809e-01)   Acc@1  79.69 ( 76.39)   Acc@5  93.75 ( 93.20)
Test: [150/391] Time  1.004 ( 0.710)   Loss 7.4515e-01 (8.9780e-01)   Acc@1  75.00 ( 76.70)   Acc@5  94.53 ( 93.49)
Test: [200/391] Time  0.120 ( 0.698)   Loss 1.1605e+00 (1.0248e+00)   Acc@1  72.66 ( 74.11)   Acc@5  92.19 ( 91.86)
Test: [250/391] Time  0.795 ( 0.696)   Loss 8.2321e-01 (1.0965e+00)   Acc@1  77.34 ( 72.67)   Acc@5  92.19 ( 90.83)
Test: [300/391] Time  0.858 ( 0.692)   Loss 1.2221e+00 (1.1582e+00)   Acc@1  76.56 ( 71.52)   Acc@5  89.84 ( 89.99)
Test: [350/391] Time  0.450 ( 0.688)   Loss 1.2706e+00 (1.2056e+00)   Acc@1  67.19 ( 70.51)   Acc@5  86.72 ( 89.31)
 * Time 0.687 Loss 1.204 Acc@1 70.461 Acc@5 89.402
Testing time: 268.5448255710071
```

(c)

Figure 15: (a) Code for Conv2d module with all sources of quantization, proposed W-reshape and A-shift implemented. (b) Code for top-level training with the proposed BitAug implemented. (c) Example training log for MobileNetV2.

6-bit ADC and with BitAug applied. Compared to Fig. 4d where BitAug is not present, we can see a reduced number of local minimum.

Figure 16: (a) Illustration of $g(\cdot)$. (b-c) Weight distribution collected at the last iteration without and with BitAug applied, respectively. (d-e) Loss surfaces starting from pre-trained checkpoints without and with BitAug applied, separately. (f) Loss surface of a 6-bit ADC with BitAug applied.

## G  COMPARISON OF QAT METHODS

Table 10 shows a comparison between different QAT methods based on the BERT-base model with 4-bit weights and activations, and 8-bit ADCs. The first column shows the name of each tested QAT method. The resting columns show the accuracy without ADC present, with ADC present but without RAOQ techniques, with ADC present and with RAOQ applied, respectively. As seen, RAOQ demonstrates a stable behavior on all of these methods, significantly restoring their performance to baseline level.

Table 10: Comparison of different QAT methods.

| Methods | QAT without ADC | QAT only | RAOQ |
|---|---|---|---|
| LSQ+ Bhalgat et al. (2020) | 87.75 | 82.43 | 87.67 |
| LSQ Esser et al. (2019) | 87.60 | 82.02 | 87.41 |
| PACT+SAWB Choi et al. (2019) | 87.49 | 80.88 | 87.34 |

## H  FRAMEWORK

Algorithm 1 summarizes our training framework, including the proposed A-shift, W-reshape, and BitAug approaches. All implementations are done in PyTorch. Once the model is trained, it is employed for inference using IMC, as shown in Fig. 17. During inference, we first transfer the input data and model parameters to the IMC system. The output is then collected and sent back to the host for further processing.

**Algorithm 1** Training framework for RAOQ. $J$ is the number of layers, $Q_X(\cdot)$ and $Q_W(\cdot)$ are conventional quantizers for activations and weights respectively, $Q_A(\cdot)$ is the ADC quantizer defined in Eq. 2, $I_Q$ and $I_A$ denote the total number of iterations for QAT and ADC phases separately.

---

**Require:** pre-trained floating-point model, input $x$
  {**Phase 1 (QAT)**}
  **for** $i = 1$ to $I_Q$ **do**
    $\mathcal{L}_\kappa = 0$
    **for** $j = 1$ to $J$ **do**
      $\overline{x} \leftarrow Q_X(x, b_x)$
      $\overline{w} \leftarrow Q_W(w, b_w)$
      $y \leftarrow \text{MVM}(\overline{x}, \overline{w})$
      $\mathcal{L}_\kappa \leftarrow \mathcal{L}_\kappa + \kappa(\overline{w})$      # $\kappa$ is defined in Eq. 4 for W-reshape
    **end for**
    ▷ Compute cross-entropy loss $\mathcal{L}_c$
    $\mathcal{L}_Q \leftarrow \mathcal{L}_c + \lambda_\kappa \mathcal{L}_\kappa$
    ▷ Backprop based on $\mathcal{L}_Q$ and update model parameters
  **end for**
  ▷ Collect updated model parameters
  {**Phase 2 (ADC)**}
  **for** $i = 1$ to $I_A$ **do**
    **for** $j = 1$ to $J$ **do**
      $\overline{x} \leftarrow Q_X(x, b_x) - 2^{b_x - 1}$      # for A-shift
      $\overline{w} \leftarrow Q_W(w, b_w)$
      $y \leftarrow \text{IMC}(\overline{x}, \overline{w})$      # IMC($\cdot$) indicates performing computation in IMC systems
      $\overline{y} \leftarrow Q_A(y, b_a)$
    **end for**
    ▷ Compute cross-entropy loss $\mathcal{L}_c(b_a)$ based on $b_a$ and backprop
    ▷ Sample $\widetilde{b}_a$ from candidate set $\mathbb{B}$      # prepare for BitAug
    **for** $j = 1$ to $J$ **do**
      $\overline{x} \leftarrow Q_X(x, b_x) - 2^{b_x - 1}$
      $\overline{w} \leftarrow Q_W(w, b_w)$
      $y \leftarrow \text{IMC}(\overline{x}, \overline{w})$      # IMC($\cdot$) indicates performing computation in IMC systems
      $\overline{y} \leftarrow Q_A(y, \widetilde{b}_a)$
    **end for**
    ▷ Compute cross-entropy loss $\mathcal{L}_c(\widetilde{b}_a)$ based on $\widetilde{b}_a$ and backprop
    $\mathcal{L}_A \leftarrow \mathcal{L}_c(b_a) + \lambda_b \mathcal{L}_c(\widetilde{b}_a)$
    ▷ Accumulate gradients from BitAug and update model parameters (refer eq...)
  **end for**

---

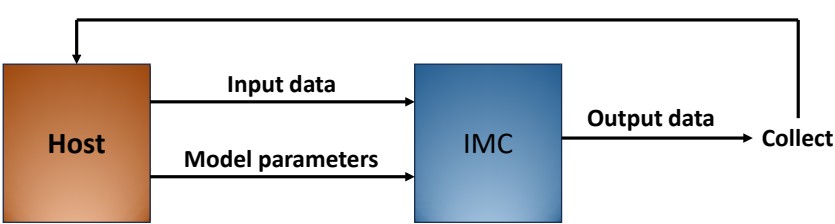

Figure 17: Inference flow using IMC.

