# OpenReview forum: "Reshape and Adapt for Output Quantization (RAOQ): Quantization-aware Training for In-memory Computing Systems"
_ICLR.cc/2024/Conference — Submitted to ICLR 2024_

### Official Review · Reviewer_5va5 · 2023-10-28

**Soundness:** 3 good
**Presentation:** 3 good
**Contribution:** 3 good
**Rating:** 6
**Confidence:** 4

**Summary:**

This paper identifies the challenge of ADC quantization error on deploying deep neural network models on IMC hardwares. The paper proposes three techniques: activation shifting, weight reshaping, and bit-augmentation to resolve the issue.

**Strengths:**

1. This paper studies a novel problem: ADC quantization error, which has not been studied before
2. The proposed methods are technically sound
3. The paper conduct experiments on different models and tasks to show the effectiveness of the proposed method, and provides ablation study for each technique.

**Weaknesses:**

1. Though the paper attempts to explain the setting of ADC quantization error in Section 3, there's still someting unclear, which makes me doubt the necessity of some proposed method. See questions below for details.
2. The proposed weight reshaping aims to improve the variance of the weight, but opt to use a complicated 4-th order regularization in Equ. (4). It is unclear why such regularization term is selected, and how is it comparing to regularizing the variance directly. Also the choice of regularization strength may need more study

**Questions:**

My main question is on the setting of ADC quantization problem. Is the hardware specification known at the training time of the model? If the specification is known, why not directly perform a post-activation shift/scaling before ADC to maximize the usage of ADC bins?

While if the hardware specification is unknown, the proposed method assumes a situation of underflow, but the degree of underflow may be unknown without hardware specification. This will make it hard to decide the regularization strength and shifting beforehand.

---

> ### Author Response · Authors · 2023-11-17
> **Response to reviewer 5va5**
>
> We appreciate the insightful feedback and suggestions provided by the reviewer. We have attempted to address all comments and suggestions in our updated manuscript. A point-by-point summary is as follows:
>
> - **W-reshape: 2nd order vs. 4th order.** This is a great question. First, the use of the 4th-order kurtosis regularization is inspired by the prior work (Shkolnik et al., 2020) which has shown that such regularization is able to flatten the weight distributions. We leverage such observation, taking it one step further by encouraging the distributions to go beyond flattening to increase the weight variance and to be adjusted by both weight and quantization parameters. Moreover, we actually tested regularization directly on the variance, but found it is not as effective as the kurtosis loss and is harder to control (i.e., high sensitivity to the strength). We show a table below for a comparison (based on the accuracy of MobileNetv2) between the regularization of variance and kurtosis. We can see that kurtosis regularization demonstrates better and more stable behavior.
> | Regularization strength  | 0.00001  | 0.00004  | 0.0008  | 0.002 |
> |--------------------------|----------|----------|---------|-------|
> | Variance                 | 69.57    | 69.62    | 65.58   | 56.25 |
> | Kurtosis                 | 69.94    | 70.02    | 70.24   | 69.65 |
>
> - **W-reshape: study on its strength.** We now include more studies of the impact on the strength of W-reshape in Appendix A. The strength of W-reshape is a hyperparameter, and we encourage users to fine-tune it for different models and applications for better accuracy.
>
> - **Hardware specification.** We do need the hardware specifications for the memory dimension, bit-precisions of activations, weights, and ADCs for a target IMC application. Ideally, we would like to perform dynamic scaling/shifting before ADC quantization, but the analog signal processing nature of IMC prevents this, as analog signals directly feed into the ADC, without power-costly variable-gain amplifiers. This is precisely what makes the algorithmic setting of ADC quantization so challenging and distinct from previous weight/activation quantization settings. Thanks for the question.

---

> ### Author Response · Authors · 2023-11-22
> **Last day reminder and further discussion**
>
> Dear reviewer 5va5,
> &nbsp;
>
> We thank again for your thoughtful and valuable comments and suggestions! As the deadline of the discussion period approaches the end, we look forward to hearing your feedback on whether the previous points of concern have now been addressed. As mentioned in our responses, we believe all points have been addressed together with manuscript revisions, and a point-by-point summary is provided. Please do let us know if you have any further questions.
> &nbsp;
>
> Best regards,
>
> Authors

---

### Official Review · Reviewer_7qFb · 2023-10-31

**Soundness:** 3 good
**Presentation:** 3 good
**Contribution:** 2 fair
**Rating:** 5
**Confidence:** 4

**Summary:**

This work proposes a Reshape and Adapt for Output Quantization (RAOQ) method to overcome ADC quantization error. This method includes activation-shifting and bit augmentation schemes. RAOQ is validated on various bit precisions and different scales of NN models for image classification, object detection, and NLP tasks, and achieves SOTA accuracy with practical IMC implementations.

**Strengths:**

1.	The paper proposes a Reshape and Adapt for Output Quantization (RAOQ) method to overcome ADC quantization error.
2.	Experimental results show that RAOQ is effective for object detection and NLP tasks, and achieves SOTA accuracy with practical IMC implementations.

**Weaknesses:**

1.	The paper does not provide a detailed analysis of the computational cost and memory requirements of the proposed method, which could be important factors to consider in practical applications.
2.	Section 6 only shows the energy and TOPS/W for different ADC bit precisions. It would be more meaningful if the power consumption and processing speed for different NN models is provided.
3.	The paper does not provide a framework for the proposed method.

**Questions:**

Please refer to weaknesses.

---

> ### Author Response · Authors · 2023-11-17
> **Response to reviewer 7qFb**
>
> Thank you very much for your review and suggestions to improve the quality of our manuscript. We have attempted to address all your concerns and suggestions in our manuscript, and provided a point-by-point summary below:
>
> - **Cost.** Based on our experiments, A-shift and W-reshape add negligible overhead during training. BitAug adds extra computations (i.e., computation and gradient accumulation of other ADC bit precisions), requiring 14% more GPU memory in the last training phase. Considering the challenging ADC quantization problem, we believe such overhead is justified to substantially restore accuracy performance. We now mention the training overhead in our updated manuscript. We would also like to emphasize that our target is to realize efficient inference on IMC applications, which are not affected by this training overhead.
>
> - **Power consumption and processing speed.** The plot in Section 6 is at the macro level, i.e., for one MVM computation. However, the power consumption and processing speed for end-to-end execution of NN models depend on the specific hardware architecture and computation mapping employed. This can not be quantified generally, and we leave such hardware architectural work out of the scope of this work.
>
> - **Framework.** We now provide the summary of our training framework as well as the flow of running IMC inference in Appendix H in our updated manuscript. Thank you for pointing it out. As mentioned in Section 5, all of the proposed methods are implemented in PyTorch. We also attach the example code in supplementary material for your reference.

---

> ### Author Response · Authors · 2023-11-22
> **Last day reminder and further discussion**
>
> Dear reviewer 7qFb,
> &nbsp;
>
> We thank again for your thoughtful and valuable comments and suggestions! As the deadline of the discussion period approaches the end, we look forward to hearing your feedback on whether the previous points of concern have now been addressed. As mentioned in our responses, we believe all points have been addressed together with manuscript revisions, and a point-by-point summary is provided. Please do let us know if you have any further questions.
> &nbsp;
>
> Best regards,
>
> Authors

---

### Official Review · Reviewer_dnyX · 2023-11-02

**Soundness:** 3 good
**Presentation:** 3 good
**Contribution:** 2 fair
**Rating:** 5
**Confidence:** 4

**Summary:**

The Reshape and Adapt for Output Quantization (RAOQ) is proposed to mitigate the ADC quantization noise in in-memory computing (IMC). The method reshapes the statistics of activations and weights and retrains the networks. It is shown that RAOQ improves the accuracy for various network models for a diverse set of tasks (image classification, object detection, NLP).

**Strengths:**

1. The proposed method extends previous works on QAT to incorporate output quantization in addition to top of weight and activation quantization. In this way, it 'completes the picture'.
2. Though the work is empirical it is supported by extensive simulation results for a variety of tasks.
3. The paper describes the work clearly.

**Weaknesses:**

1. Application of this work is limited to in-memory computing. This is a niche area. The work would be more useful if it could be applied to digital computation as well.
2. The paper would be a stronger if it presented itself as an advanced form of QAT, one that includes output quantization.
3. Comparison with multiple QAT approaches is missing. Only one [Bhalgat] is considered.

**Questions:**

1. Fig. 5 compares energy efficiency of IMC vs. digital accelerators. Was this comparison done at iso-computational or network accuracy, e.g., did both architectures display the same misclassification rate for image classification?
2. What is the impact of RAOQ on weight and activation sparsity? Does it increase or decrease it or is there no effect? If sparsity is reduced then it will affect complexity reduction techniques that rely on it.
3. What is the training overhead due to RAOQ? For large AI models, even if training is done once or occasionally, it is already complex. RAOQ will make it even more complicated.
4. Table 1 (main result) is missing error bars. It is hard to say if the results are statistically significant.
5. How does RAOQ perform when subject to out-of-distribution (OOD) samples during inference? This is a very practical problem with BP-based training in general.
6. Table 3 caption has a typo.

---

> ### Author Response · Authors · 2023-11-17
> **Response to reviewer dnyX**
>
> We thank the reviewer for the thoughtful review and suggestions to improve our manuscript. We have attempted to address all comments and suggestions in our updated manuscript. A point-by-point summary is as follows:
>
> - **Applicability and extension of QAT.** We thank the reviewer for bringing up the good point. Technically, our proposed approaches fall into the class of QAT methods, and are applicable to any systems involving output quantization. Analog IMC is one of the most widely researched approaches to addressing compute energy of neural-network inference; however, output quantization is especially challenging since the ADC quantization step is fixed, i.e., no learnable quantization parameters. Both due to its wide interest and the impact of output quantization, we frame this work in the context of IMC. Comparatively, the output quantization of digital systems is typically much more modest (i.e., either with higher bit-precision available or with learnable step size) compared to that in analog IMC.
>
> - **Other QAT methods.** We now include a comparison between different QAT methods in Appendix G. As we can see, different QAT methods behave similarly in terms of ADC quantization and our methods are applicable to all of them.
>
> - **IMC vs. digital accelerators.** Fig. 5 is just showing an energy comparison between IMC and a typical digital accelerator architecture. The motivation of this work is that, while IMC achieves higher efficiency, its accuracy is typically degraded due to ADC quantization. Thus, in this work, we aim to restore the degraded performance to iso-accuracy.
>
> - **Sparsity.** The table below shows the sparsity for example models with and without applying RAOQ. As we can see, the proposed RAOQ does not affect the sparsity level.
> |   Methods | ResNet50  | MobileNetv2 | YOLOv5s | BERT-base |
> |--------------|-----------|-------------|---------|-----------|
> | Without RAOQ | 1.15%     | 0.88%       | 0.89%   | 1.11%     |
> | With RAOQ    | 1.20%     | 0.79%       | 1.01%   | 1.08%     |
> - **Training overhead.** Based on our experiments, A-shift and W-reshape introduce negligible training overhead. BitAug does require 14% more GPU memory and slows down the training process by 1.5$\times$ for the last training phase, due to the computation and accumulation of gradients associated with different ADC bit precisions. We propose that such training overhead is an incurred cost for improving the accuracy as shown in Table 1. We now include the training overhead in our updated manuscript.
> - **Error bars.** The results shown are averaged from 3 runs, which is observed to provide a statistically stable behavior. We kept the table compact without error bars for clarity. But, below we show some examples with error bars for 4-bit weights and activations with 8-bit ADCs, for your reference:
> |   &nbsp;       | ResNet50          | MobileNetv2     | BERT-base         | YOLOv5s         |
> |----------|-------------------|-----------------|-------------------|-----------------|
> | Accuracy | 76.27$\pm$0.05   | 70.46$\pm$0.07 | 87.67$\pm$0.07  | 33.49$\pm$0.03 |
>
> - **Out of distribution samples.** RAOQ does not modify the way the input is quantized, thus does not have a direct impact on OOD. Detailed analysis of OOD samples is out of the scope of this work, where we instead align to standard evaluation methodologies employed for QAT studies. We will consider the impact of OOD samples in our future studies. We thank the reviewer for the suggestion.
>
> - **Typo.** We have fixed the typo in our updated manuscript. Thanks for pointing out.

---

> ### Author Response · Authors · 2023-11-22
> **Last day reminder and further discussion**
>
> Dear reviewer dnyX,
> &nbsp;
>
> We thank again for your thoughtful and valuable comments and suggestions! As the deadline of the discussion period approaches the end, we look forward to hearing your feedback on whether the previous points of concern have now been addressed. As mentioned in our responses, we believe all points have been addressed together with manuscript revisions, and a point-by-point summary is provided. Please do let us know if you have any further questions.
> &nbsp;
>
> Best regards,
>
> Authors

---

### Official Review · Reviewer_zm7p · 2023-11-05

**Soundness:** 2 fair
**Presentation:** 3 good
**Contribution:** 3 good
**Rating:** 5
**Confidence:** 5

**Summary:**

This paper addresses the issue of accuracy degradation due to ADC quantization in In-Memory Computing (IMC) and proposes a solution called RAOQ. RAOQ introduces three techniques to adjust the statistical properties of weights and activations, as well as to enhance the model optimization process. Firstly, W-reshape adjusts the statistical distribution of the weights to maximize the Signal to Quantization Noise Ratio (SQNR). Secondly, A-shift modifies the range of activation values to minimize the impact of ADC quantization. Lastly, BitAugs leverages ADCs with various bit precisions to improve model optimization. The performance of RAOQ has been evaluated across various datasets, models, and bit precisions, consistently achieving high accuracy.

**Strengths:**

- This paper attempts to solve an important challenge of ADC precision as ADC complexity becomes a major performance bottleneck for IMC.

- This paper assesses the performance of the proposed algorithm across a wide variety of models and tasks to evaluate its generality, demonstrating consistent performance improvements.

**Weaknesses:**

- Reproducibility issue: the paper discusses ADC quantization error, a new type of error that reflects the non-ideal property of analog signal processing; but modeling such error on DNN training is not trivial. Furthermore, the training procedure includes complex tuning of regularization and augmentation, which would also hinder the reproduction of the claimed benefits. Thus, to reproduce the impact of ADC quantization on the models and the effectiveness of RAOQ, it is desirable to provide a reference code that reveals the error modeling of ADC quantization and the hyper-parameter settings.

- Lack of fundamental understanding of distribution shifts on SQNR. The authors claim that increasing the variance of W and the 2nd moment of X would improve SQNR. However, there is only limited empirical evidence to support their claims without theoretical justification. Therefore, there are many unanswered questions such as:
1) They ignore the impact of the distribution shift of W and X on quantization errors.
2) The claim that Var[Y] can be maximized by maximizing Var[W] and E[X^2] is weakly supported.
3) Fig. 3.b shows a compound effect of W-reshape and A-shift, but does not show an individual impact separately.

- The rationale for enhancing performance by adding the loss terms for various bit precisions in BitAug to the original loss and gradient calculations seems insufficiently explained. The authors properly showed the difficulty of fine-tuning with ADC quantization, but there is little justification for why the proposed BitAug helps optimization.

- The bit precision candidate selection in BitAug is empirical. According to Appendix C, the bit
precision candidate set that shows good performance in MobileNetV2 is applied across all other
networks, including BERT and ResNet

- As the authors admitted, the Kurtosis regularization for weight quantization was already proposed by (Shkolnik et al., 2020) with a similar purpose of flattening weight distribution. Therefore, the novelty of W-reshape is not clear.

- Not all operations of the model are mapped to IMC. Based on Appendix E, while the research
argues that the operations not mapped to IMC (e.g., the first and last layer in CNN, depth wise
convolution in the MobileNet family, the second matrix matrix multipl ication inside the Multi Head
Attention of BERT) represent a small fraction of the total operations, these operations are typically
located intermittently within models. This may induce overhead (e.g., data transfer latency between
the host and IMC

**Questions:**

- What is the precision of the short-cut (for ResNet, MobileNet, BERT)? (Since the output of MatMul/Conv is taken to the short-cut in pre-activation residual, ADC quantization would affect short-cut precision as well.)

- Regarding A-Shift, if enlarging E[X^2] is crucial, why don't we shift X more aggressively toward negative? (instead of stopping at zero-center, in Fig. 3.a(bottom))

- For the same ADC bit-precision, how can BitAug adjust the loss surface? (e.g., can you compare loss surface of 6-bit ADC in Fig. 4 without and with BitAug?)

---

> ### Author Response · Authors · 2023-11-17
> **Response to reviewer zm7p (1/2)**
>
> We thank the reviewer for the review and thoughtful suggestions to improve our manuscript. We have attempted to address all comments and suggestions in our updated manuscript. A point-by-point summary is as follows:
>
> - **Reproducibility.** The modeling of ADC quantization has been done routinely in signal processing systems and is quite well understood. In fact, it follows quantization methods that have been used in DSP and deep-learning weight/activation quantization, and can be accurately modeled (Eq. 2). While RAOQ introduces new hyperparameters (i.e., $\lambda_A$ and $\lambda_\kappa$) like other QAT methods (e.g., [1-3]), they are straightforward to adjust. A global setting of $\lambda_A$ can be applied across all models as mentioned in Appendix E, and $\lambda_\kappa$ can be set by analyzing the quantized weight distributions as studied in Appendix A. To demonstrate and validate the effectiveness of RAOQ, we have attached example code for ResNet50 and MobileNetv2 as supplementary material for your reference, and will release more models along with publication.
>
> - **Effect of distribution shift of W and X on quantization errors.** Both A-shift and W-reshape are aimed at adjusting the output distribution to mitigate the effects of output (ADC) quantization. A-shift does not introduce any additional quantization errors on the activation quantization itself. The impact of W-reshape was studied and included in Appendix A.
>
> - **Support that var[Y] can be maximized by maximizing Var[W] and E[$X^2$].** We agree with the reviewer regarding the weak support of the formal claim around maximization - we did not intend to make the formal claim and have corrected it. Rather, the argument relies on analytical intuition and empirical validation. Analytically, such a relationship is clearly seen before training starts as weights and activations are independent of each other. As training proceeds, a complex correlation between weights and activations arises, along with a correlation between the elements of the activation tensors. This makes it difficult to derive a formal analytical maximization criterion. Instead, we seek to show empirically through the provided data (Fig. 2(c-d)) that the analytical intuition in terms of their proportionality continues to hold through training, and provides simple yet effective ways (i.e., W-reshape and A-shift) to increase Var[Y] to improve SQNR in the presence of ADC quantization. We have clarified this in the updated manuscript.
>
> - **Individual impact of A-shift and W-reshape.** Thank you for pointing this out. We have modified Fig. 3b to include both their compound effect and their individual effects. As we can see, both of them contribute to the SQNR improvement.
>
> - **Justify BitAug and impact on loss landscape.** We argue that BitAug assists the learning process by avoiding getting stuck at local minima and potentially maintaining a smoother loss landscape. We now include more details in Appendix F to show our analysis for justification and provide the requested loss surface after applying BitAug for 6-bit ADC.
>
> - **BitAug (selection of candidates).** We agree that the selection of these candidates is empirical. But, the approach is to employ a narrow range around the target ADC precision, to optimize to this target, while employing high enough precision to assist in training and low enough precision to ensure adequate parameter adaptation to quantization errors. We would like to emphasize that, unlike weight/activation quantization, ADC quantization does not provide for any learnable quantization parameters, thus making the learning process more difficult. Hence, providing guidance with very different learning objectives would not help much (shown in Table 6). Thus, we mainly focus on the candidates around the target ADC precision. We provide more evidence to justify our arguments in Appendix C.
>
> **References**:\
> [1] Choi, Jungwook et al. “Accurate and Efficient 2-bit Quantized Neural Networks.” Conference on Machine Learning and Systems (2019).\
> [2] Gong, Ruihao et al. “Differentiable Soft Quantization: Bridging Full-Precision and Low-Bit Neural Networks.” 2019 IEEE/CVF International Conference on Computer Vision (ICCV) (2019): 4851-4860.\
> [3] Nagel, Markus et al. “Overcoming Oscillations in Quantization-Aware Training.” International Conference on Machine Learning (2022).

---

> ### Author Response · Authors · 2023-11-17
> **Response to reviewer zm7p (2/2)**
>
> - **Novelty on W-reshape.** The target of this work is to develop methods to improve model accuracy with ADC quantization, and our use of W-reshape to realize this is distinct. Our work leverages the observation in (Shkolnik et al., 2020) that kurtosis regularization can be used to flatten the weight distribution, which was aimed to improve the weight quantization robustness. But, both the use and objective of such regularization are very different in our work. First, we are interested in the distribution of quantized weights rather than the floating-point weights. By applying kurtosis regularization on quantized weights, we add one more degree of freedom, allowing both weight parameters and quantization parameters to adapt to the desired distribution. Second, flattening (i.e. uniformly distributed) weights are not necessarily the best choice for output quantization. Thus, we push further, allowing weight distributions to be even endpoints-concentrated (e.g., Fig. 3(a)), specifically to benefit ADC quantization.
>
> - **Overhead from layers not mapped to IMC.** The typically envisioned architecture of IMC accelerators involves also using proximate digital acceleration units to handle layers that are not mapped to IMC (e.g., [1]). This avoids communication overheads to a host processor.
>
> - **Short-cut precision.** Short-cut connections are quantized and mapped to IMC in the same way as other layers.
>
> - **Amount of shift in A-shift.** Great question. The amount of shift is limited by the minimum/maximum value that the quantized activation range can support. We have already maximized the shift within this limit.
>
> - **How can BitAug adjust the loss surface.** Please refer to our main reply to **Justify BitAug and impact on loss landscape**. We have plotted the loss surfaces for 6-bit ADC per your request in Fig. 4 (without BitAug) and Fig. 16 (with BitAug).
>
> **References**:\
> [1] K. Ueyoshi et al., "DIANA: An End-to-End Energy-Efficient Digital and ANAlog Hybrid Neural Network SoC," 2022 IEEE International Solid-State Circuits Conference (ISSCC), San Francisco, CA, USA, 2022, pp. 1-3

---

> ### Author Response · Authors · 2023-11-22
> **Last day reminder and further discussion**
>
> Dear reviewer zm7p,
> &nbsp;
>
> We thank again for your thoughtful and valuable comments and suggestions! As the deadline of the discussion period approaches the end, we look forward to hearing your feedback on whether the previous points of concern have now been addressed. As mentioned in our responses, we believe all points have been addressed together with manuscript revisions, and a point-by-point summary is provided. Please do let us know if you have any further questions.
> &nbsp;
>
> Best regards,
>
> Authors

---

### Meta-Review · Area_Chair_2QV1 · 2023-12-10

**Metareview:**

The paper received 4 reviews. Three reviewers suggested that the paper is below the acceptance bar. Reviewer 5va5, who placed the paper slightly above the bar, shared several concerns about clarity, hence, their score cannot overweight the concerns of other reviewers. The AC, therefore, decided to reject the manuscript.

**Justification For Why Not Higher Score:**

Three reviewers suggested rejection, forth reviewer shared clarity concerns and provided marginally above score.

**Justification For Why Not Lower Score:**

N/A

---

### Decision · Program_Chairs · 2024-01-16

Reject